

# Unbiased statistical length analysis of linear features: Adapting survival analysis to geological applications

Gabriele Benedetti[1], Stefano Casiraghi[1], Daniela Bertacchi[2], and Andrea Bistacchi[1]

[1]Dipartimento di Scienze dell'Ambiente e della Terra, Università degli Studi di Milano-Bicocca, 20126 Italy
[2]Dipartimento di Matematica e Applicazioni, Università degli Studi di Milano-Bicocca, 20125, Italy

**Correspondence:** Gabriele Benedetti (gabriele.benedetti@unimib.it)

**Abstract.** A proper quantitative statistical characterization of fracture length (or height) is of paramount importance when analysing outcrops of fractured rocks. Past literature suggested adopting a non-parametric approach, using circular scanlines, for the unbiased estimation of the fracture length mean value. However, necessities shifted and now there is an increasing demand for parametric solutions to correctly estimate and compare all the parameters (e.g. mean AND standard deviation) of several types of distributions. These changing requirements highlighted the absence in geological literature of properly

structured theoretical works on this topic and in particular on different biases that affect this estimate. Here we propose to tackle the right censoring bias, caused by limited size of outcrops with respect to fracture length, by applying survival analysis techniques: a branch of statistics focused on modelling time to event data and correctly estimating model parameters with data affected by censoring. After discussing both theoretical and practical aspects of survival analysis applied to geological

datasets, we propose a novel approach for selecting the most representative parametric model (i.e. statistical distribution), combining a direct visual approach and distance statistics modified to accommodate for censored data. The proposed approach has been applied to real outcrop data, correctly estimating censored length distributions. We also show the effects of censoring percentage on crude parametrical estimation that do not use this paradigm. The theory and techniques discussed here are wrapped in an easily installable open-source Python package called FracAbility (https://github.com/gecos-lab/FracAbility).

## 15  1  Introduction

Fractured rock masses are complex systems composed by intact rock and discontinuities (Hoek, 1983). Characterizing the statistical distribution of 3D geometrical properties of such discontinuities (e.g. aperture, roughness, area, orientation, height/length ratio, etc.) is fundamental for understanding and modelling mechanical and hydraulic properties of rock masses and fluid-rock interaction. Nowadays the increase in computing power and new approaches based on Digital Outcrop Models (DOMs) (Bis-

tacchi et al., 2015; Tavani et al., 2016; Bistacchi et al., 2020; Martinelli et al., 2020; Bistacchi et al., 2022; Mittempergher and Bistacchi, 2022) allows the extraction of large datasets and facilitates the measurement of properties instead of just their estimation (Marrett et al., 2018; Storti et al., 2022). The need for a more rigorous statistical approach of structural data analysis is also motivated by the popularization of stochastic Discrete Fracture Networks (DFNs) as a modelling approach for rock masses. In DFNs, discontinuities are represented, in a simplified way, as finite planar surfaces, generally rectangular, polyg-



onal or elliptical (Cacas et al., 1990; Dershowitz et al., 1992; Tavakkoli et al., 2009; Hyman et al., 2015). In fact, stochastic algorithms used to generate these surfaces are guided by statistical distributions obtained from field or well data, or assumed based on some prior knowledge (Andersson et al., 1984; Cacas et al., 1990; Davy et al., 2018). For example, DFNs where fractures are represented as rectangular surfaces (the most common implementation) require defining fracture size via parametric distributions of length (measured along strike) and height (measured along dip), or alternatively length and length/height ratio
(Hyman et al., 2015).

The main obstacle that geologists encounter while trying to characterize a fractured rock volume is the impossibility of directly measuring the 3D properties of discontinuities, since only indirect geophysical methods may provide truly 3D datasets. However, imaging discontinuities with geophysical methods is strongly limited by spatial resolution and/or by the absence of contrast in the physical properties investigated by a particular technique (Martinelli et al., 2020). Because of this, a rich
literature has been developed focusing on the characterization of discontinuity traces or lineaments, i.e. the 2D lines of intersection of 3D discontinuity surfaces with the outcrop surface, or with topography, with the goal of measuring orientation, length, fracture density and intensity (Dershowitz and Herda, 1992), spacing (Bonnet et al., 2001; Storti et al., 2022), topology (Sanderson and Nixon, 2015; Manzocchi, 2002), roughness (Bistacchi et al., 2011), aperture (Bonnet et al., 2001), fracture spatial distribution (Bistacchi et al., 2020) and representative elementary volume (Martinelli et al., 2020). In this contribution
we focus on the problem of defining accurate and unbiased length (or height) distributions based on observations carried out on natural outcrops with the Digital Outcrop approach. In any given outcrop, lineament length measurements will always be affected by four main biases: length (i.e. size), censoring, truncation and orientation (Baecher, 1983). Bias correction has been thoroughly researched, and the standard solution currently adopted by many Authors is based on circular scanlines (Mauldon, 1998; Zhang and Einstein, 1998; Mauldon et al., 2001; Rohrbaugh et al., 2002; Healy et al., 2017). The method consists in
drawing a circle directly on the outcrop or on an image and counting the intersections of the circle with lineaments. With a chain of assumptions, defining an indirect relationship between mean length and the intersections of lineaments with the scanline (Mauldon et al., 2001), an unbiased non-parametric estimation of mean length, fracture density and intensity can be obtained. Thanks to its simple implementation in the field, this technique became popular and, thanks to its computational efficiency and apparent simpleness, was also implemented in modern applications such as FracPaQ (Healy et al., 2017) to be used with the
DOM approach. However, this method has an important limitation: lineament lengths are **never directly measured** and so the circular scanline method yields estimates of mean values **without** a complete characterization of the lineament length distribution and **without** any real statistical significance (e.g. variance can be estimated only under very limiting assumptions, Pahl (1981)). Despite this problem, the popularity of this method was motivated by the fact that, in the past without modern digital imaging techniques, length data acquisition was slow and tedious and thus datasets were usually small. Moreover, calculating
the length and estimating any distribution other than the exponential, was difficult and computationally intensive (Baecher and Lanney, 1978; Baecher, 1980) and, due to limitations in early algorithms used to generate stochastic fracture networks, there was no real interest in estimating precise distribution parameters.

Our methodology, presented for the first time in this contribution, represents a powerful alternative to circular scanline methods to treat specifically the censoring bias and obtain an unbiased trace length statistical model. This specific bias is defined when





for some traces, one or both ends cannot be seen due to the limited size of the outcrop (Baecher and Lanney, 1978). This effect is present from thin section to satellite image scale, and it is caused by the inability to see beyond the study area (i.e. thin-section limits, outcrop extension and so on) (Mauldon et al., 2001). From a statistical point of view, this problem is analogous to the censoring bias affecting some medical, biological, and engineering datasets, and the techniques used in these disciplines to solve or limit the effects of this bias go under the names of survival analysis, life testing, or reliability analysis (Kaplan

and Meier, 1958; Leung et al., 1997; Lawless, 2003; Cox, 2017; Karim and Islam, 2019). Even though in these disciplines the recorded random variables are **times** (e.g. lifetime of a patient, time-to-failure of a mechanical part, etc.), we demonstrate that this statistical technique can be also adapted to **lengths**. Measuring length is straightforward with dedicated code or with a simple GIS software but applying survival analysis and fitting robust parametric statistical distributions, notwithstanding these biases, needs a more detailed treatment that is the main topic of this contribution. We propose to physically measure the length

of each fracture that is digitized onto the DOM, adapt survival/reliability analysis techniques to censored trace length datasets and demonstrate the possibility of estimating robust trace length distributions from datasets that include censored data. As a second objective we propose a quantitative methodology to select the most representative estimated statistical model (i.e. parametric distribution) from a list of hypotheses. We wrapped the theory and techniques presented in this paper in an open-source Python package called FracAbility. The library works with shapefiles as input, and it allows to carry out a complete and

unbiased statistical analysis workflow for fracture length data (https://github.com/gecos-lab/FracAbility).

## 2 Fracture surveys and terminology

A discontinuity in a rock mass can be defined as surfaces across which a material has lost its cohesion, thus including faults, fractures, foliations, stylolites, compaction or deformation bands, and bedding interfaces. Fractures can be additionally subdivided into shear and tensional fractures, the latter further divided in joints when empty or veins when filled by minerals (Twiss

and Moores, 2007; Davis et al., 2012). Discontinuities that have the same formation age, kinematics and orientation can be grouped into families or sets and multiple sets form a fracture network or system (Davis et al., 2012). Fundamental for its impact on the mechanical and hydraulic properties of fractured rock mass, is the connectivity of a fracture network (Gueguen et al., 1991), that will be characterized by intersection lines where pairs of fractures show crosscutting or abutting relationships (Hancock, 1985). Although three dimensional by nature, most of the times discontinuities appear as 2D lineaments or traces,

i.e. the intersection of discontinuity surfaces with a secondary surface that reveals the internal structure of the rock mass, such as the outcrop or topographic surface (but also a surface cut in a sample or a borehole). Expanding on this basis, fracture networks can be considered, as they appear on 2D outcrops, as the composition of three main components: *Fractures*, *Boundaries* and *Nodes* (Figure 1A). In this work, following a common usage in outcrop studies, the noun term *fracture* is used to indicate any type of discontinuity trace and intersections between fracture traces with other traces are defined as *nodes* (Manzocchi,

2002; Sanderson and Nixon, 2015). In addition, for a proper statistical characterization we formally define the boundary of the sampling area (from a thin section to a satellite image), within which the sampling of fracture traces is assumed to be complete (missing some fractures in the sampling area leads to underestimating fracture density and other parameters). Furthermore,





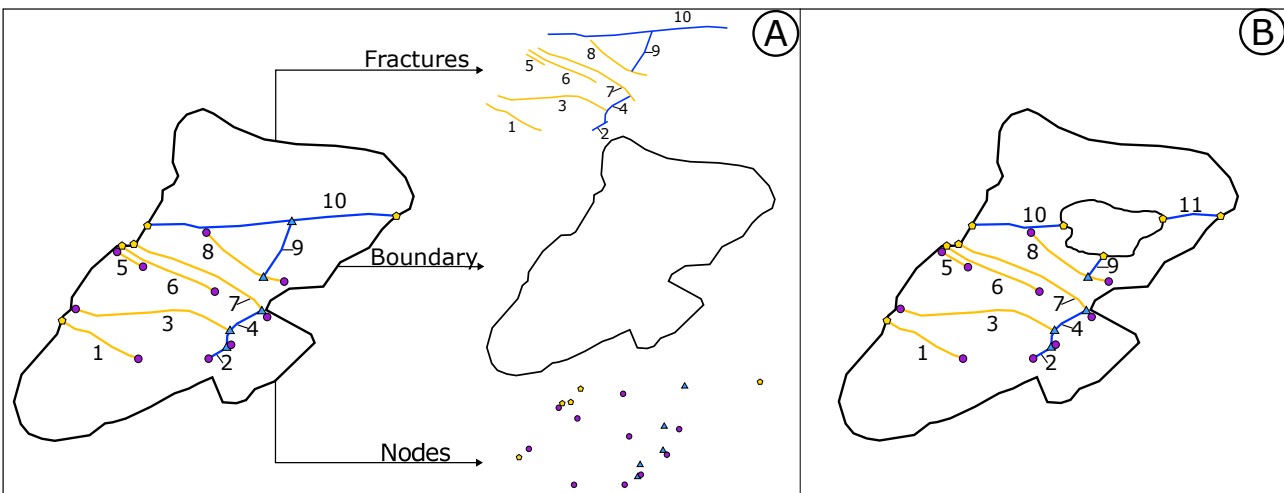

**Figure 1.** In A an example of a simple fracture network and its' components. In B, a modified version of the boundary with a "hole" in which no interpretation can be carried out. The presence of these holes can increase the number of censored fractures and introduce uncertainty on the interpretation, splitting fractures in multiple pieces (i.e. fracture 10 is split in two).

**boundary nodes** can be defined as the intersection between fracture traces and the sampling area boundary, i.e. defining censored traces. In the sampling area, also holes can be present, indicating areas within the boundary in which it is impossible to carry out the interpretation because of excessive localized alteration, anthropogenic activity, vegetation etc... The presence of these holes can increase the number of censored fractures and introduce uncertainty in the interpretation of fractures (Figure 1B).

Nodes and branches define the topology of the network (Sanderson and Nixon, 2015). Topology is a branch of mathematics used to study properties that are preserved through continuous transformations of space or scale (i.e. without tearing or ripping) (Mallet (2002, p. 28), Poincaré (2010)). Nodes can be classified by counting the number of connections to branches:

- Isolated **I** nodes are connected to one branch only.

- **Y/T** and **X** nodes are respectively connected to 3 and 4 branches.

- "Boundary" **B** nodes are located at the intersection between a fracture and the sampling area boundary (**U** nodes in Nyberg et al. (2018)).

Nodes also have important geological significance. **I** nodes indicate that fractures terminate within the observational boundary and do not end or cross with other fractures. A network with a prevalence of **I** nodes, will be less connected and fluid flow will be more restrained. **Y/T** nodes can correlate to either abutting or crosscutting relationships depending on the type of discontinuity; abutting is typical for discontinuities with no displacement, while crosscutting for discontinuities with visible displacement. **X** nodes on the other hand correlate to crosscutting relationships. Both **Y/T** and **X** nodes contribute heavily to connectivity and hence fluid flow since they increase the connectivity of the network and thus the permeability. Finally, **B**





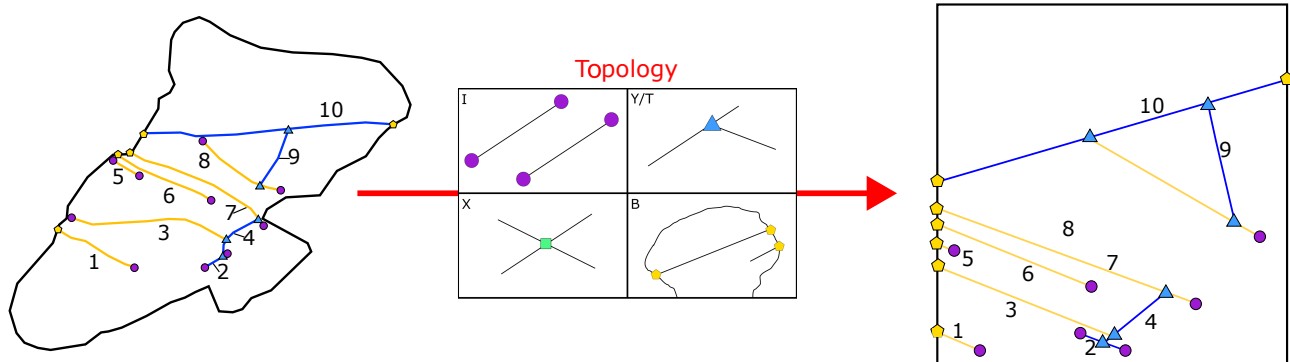

**Figure 2.** Topological abstraction of the simple fracture network in Figure 1 following the described node classification.

nodes indicate that the fracture is censored because it is not completely exposed in the sampling area and thus some measured properties are incomplete (such as length) (Figure 2).

## 3 Statistical modelling of censored length data

Having a length dataset, there is the necessity to estimate parameters of statistical distributions however, censoring is inevitable because the boundary area in which we measure the fracture trace will always be limited. Then **how can we produce unbiased estimations of parameters of one or more statistical models (such as the mean length of the fractures)?** Common and simplistic approaches such as ignoring censoring (i.e. considering censored lengths as if they were complete) or excluding censored measurements (i.e. cherry-picking only non-censored data) must be avoided at all costs as they will always lead to an underestimation of the model parameters (see discussion for a more in depth analysis). Circular scan lines methods on the other hand do offer an unbiased estimate of the mean length, however, being non-parametric, additional fundamental parameters such as the standard deviation and skewness of the distribution, cannot be estimated, making the estimate almost completely meaningless and useless in applications such as stochastic modelling. To solve these problems, we propose to use survival analysis, a specialized field of statistics that was specifically developed to deal with time of occurrence of an event of interest (Kalbfleisch and Prentice, 2002). The advantage of survival analysis with respect to methods discussed above is that considers censored data as the carrier of the crucial information that *the event did not occur up to the censoring time* and thus it properly treats censoring, allowing for an unbiased estimation of all statistical parameters and models. In this chapter, we will briefly summarize the theory of survival analysis and show how it can be used in geological applications.

### 3.1 Survival analysis theory and standard terminology

Since survival analysis is rooted in medical and biological applications, the event of interest is often defined as *death*, while a *loss* indicates that observation has been lost because the observation was hindered by a secondary event, called a censoring





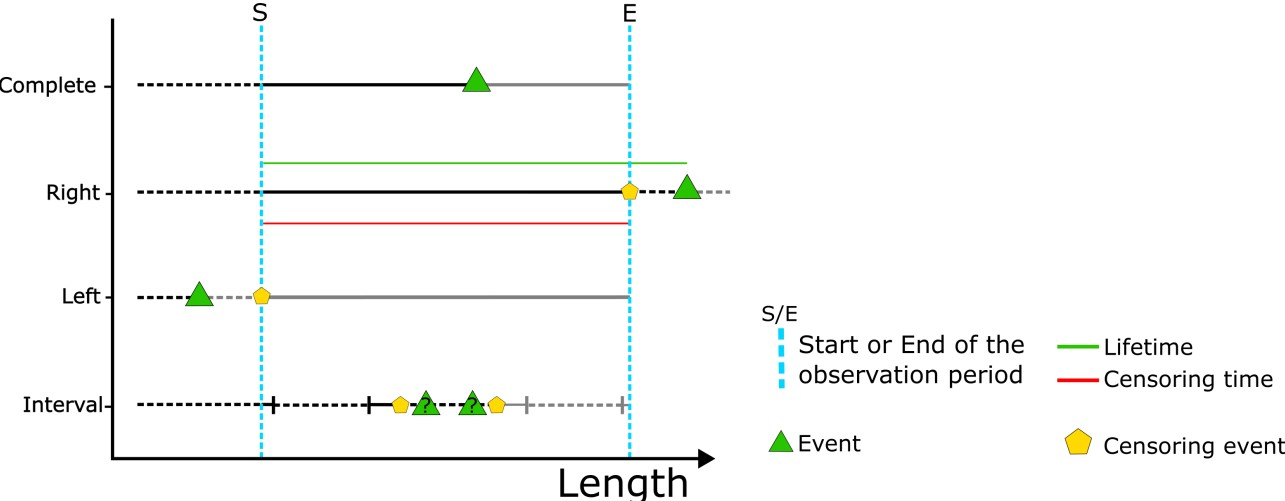

**Figure 3.** Different censoring types in a fixed observation period. Right censoring is defined when the event happens after the end of the study. Left censoring occurs when the event happens before the start of the study. Interval censoring happens somewhere between observations intervals within the study period.

event (Kaplan and Meier, 1958). Censoring events can be classified in three main types depending on when censoring happens in respect of the observation period (Karim and Islam, 2019) (Figure 3):

1. **Right censoring**: the event happens after the end of the study period and thus we partially observe the length of the event;


2. **Left censoring**: the event happens before the start of the study and thus we do not observe it;

3. **Interval censoring**: the event happens somewhere between observations intervals within the study period, usually because it is impossible to continuously monitor the occurrence of the event.

The most treated and common type is **right censoring**, that can be further divided in (Figure 4):

1. **Single censoring**: when there is an imposed end time equal for all events (i.e. *controlled* loss);


2. **Random censoring**: when each measured event is characterized by a random censoring event (i.e. *accidental* loss).

### 3.2 The survival curve and the Kaplan-Meier estimator

A fundamental assumption in survival analysis is the **independence** of the right censoring mechanisms, i.e. the assumption that the probability of occurrence of the event **does not depend** on the censoring mechanism (Kalbfleisch and Prentice, 2002; Lawless, 2003; Kleinbaum and Klein, 2012). Independent censoring is also called *non-informative censoring* as it **does not**

**affect inference** and only indicates that the time to failure exceeds the censoring time (Kalbfleisch and Prentice, 2002). This



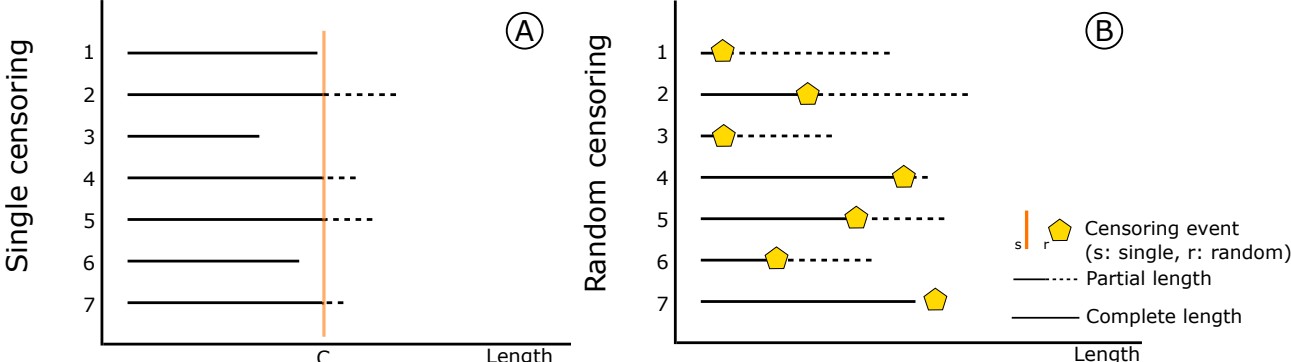

**Figure 4.** Different right censoring types for the same seven events. In A represented single censoring where events 1, 3 and 6 are complete measurements while the remaining are censored at time C. In B only event 7 is complete while the others are all censored at different times

assumption is the basis of the non-parametric Kaplan-Meier estimator of the empirical survival function (**SF**), i.e. the population probability that $X_i$ exceeds a given value $x$ (Kaplan and Meier, 1958; Kalbfleisch and Prentice, 2002):

$$SF(x) = P(X_i > x) \tag{1}$$

Often it can be useful to think the empirical survival function as the complement of the cumulative density function (**CDF**):

$$CDF(x) = P(X_i \leq x) = 1 - P(X_i > x) = 1 - SF(x) \tag{2}$$

The non-parametric Kaplan-Meier estimator $\widehat{P}(x)$ is defined by ordering N values of complete $x_{co}$ and censored $x_{ce}$ data by increasing magnitude, with $x_1 \leq x_2 \leq ... \leq x_N$, such that (Kaplan and Meier, 1958; Kalbfleisch and Prentice, 2002):

$$\widehat{P}(x) = \prod_r \frac{N - r}{N - r + 1} \tag{3}$$

where $r$ are the indexes in which the complete values are smaller than a value x (i.e. $x_{co} \leq x$).

Therefore $\widehat{P}(x)$ is a step-function that (Figure 5):

1. remains constant in any given time interval where no new events are recorded or where censoring occurs;

2. decreases, by a degree (step) depending on the number of events that occur in a time interval.

Empirical survival curves are fundamental for representing, comparing, and understanding the survival rates of different censored datasets. The curve's steepness is directly proportional to the survival rate and the median of the (random) survival time





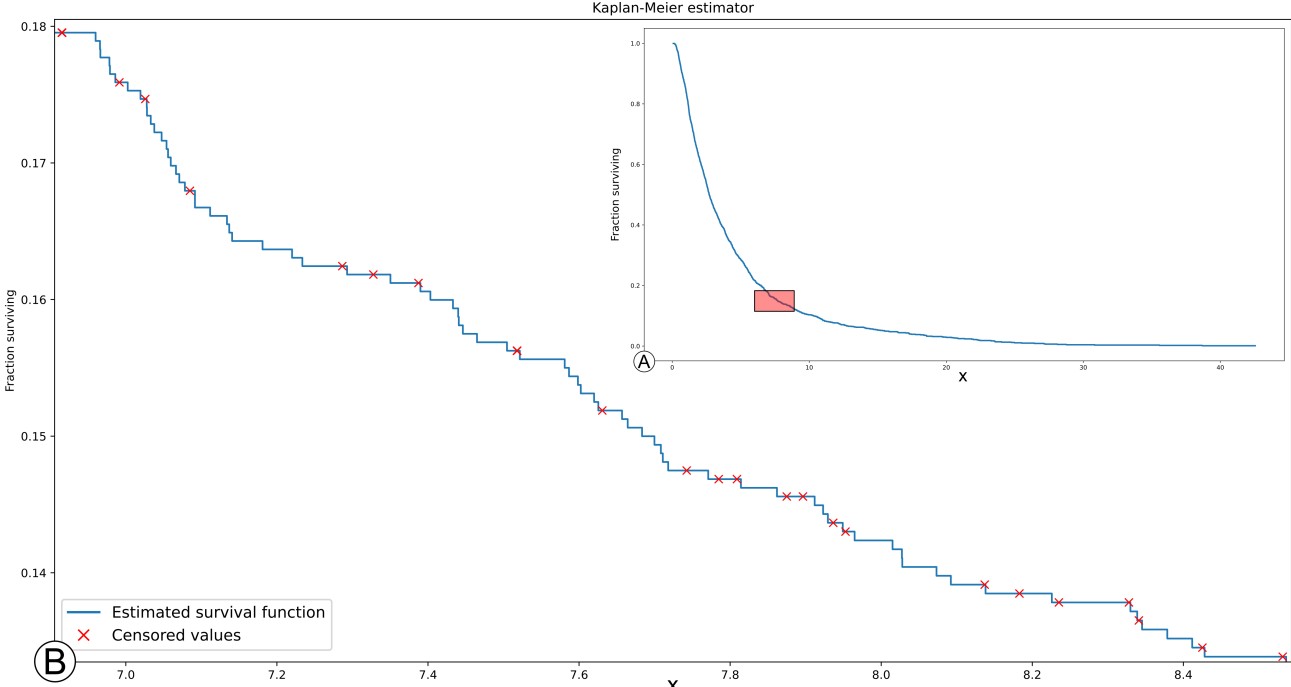

**Figure 5.** An example of the Kaplan-Meier estimated survival curve. From A the estimated curve seems continuous however by zooming in (red rectangle in A) it is a step function (B). The red Xs indicate the censoring event and show how the function never changes on these values.

(i.e. time value corresponding to a 50% chance of survival) can be used as a simple indicator to compare different survival curves.

### 3.3 The time-length dimensional shift

In literature, the term survival times, time-to-event, or more generally *lifetimes* (Lawless, 2003) seem to imply that time is the only valid variable, however any non-negative continuous variable is valid for survival/reliability analysis (Kalbfleisch and Prentice, 2002; Lawless, 2003). This silver lining is the central point of this work. Since length is, as time, a non-negative continuous variable, it is theoretically possible to apply survival analysis techniques to length datasets by considering (Figure 6):

1. the **complete** fracture trace length analogous to the **time-to-event**;

2. the **event** as the end (*death*) of a trace, marked by a node (I and/or Y/T nodes in case of complete traces);

3. the study area, defined by its boundary, analogous to the **study period**;

4. the **censored event** as the intersection between the fracture trace and the boundary (marked by a B node).





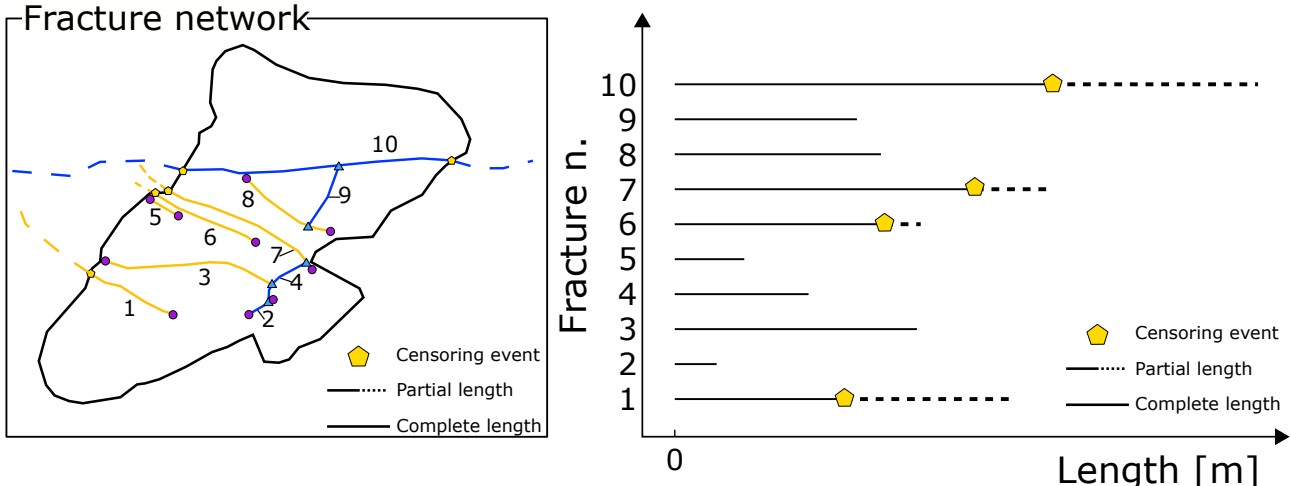

**Figure 6.** Censoring effect on an example of a simple fracture network and corresponding survival diagram.

By applying the definitions of the different types of censoring to our specific application, it is reasonable to assume that only
random censoring occurs in trace length analysis and that censoring is non-informative since the boundary is the product of
secondary events that occur after the fracture genesis (i.e. alteration, erosion, debris covering parts of the outcrop, vegetation,
human activity, etc.) and thus do not inform in any way the occurrence of the event (see the discussion for a more in-depth
analysis).

### 3.4 Unbiased MLE model estimation for censored data

Considering that the censored event **does not** modify the probability of occurrence of the measured event, survival analysts try
to solve the problem of estimating a parametric statistical model (i.e. distribution) by using optimization algorithms such as
linear regression or the Maximum Likelihood Estimator (MLE). Both methods are valid, however in these types of applications
MLE is most used and thus we will discuss more in detail this technique.

MLE is a consistent (i.e. estimation approaches the population parameter as sample size increases) and efficient (i.e. lowest
variance) estimator (Enders, 2005). The main objective of MLE is (Burnham and Anderson, 2002, 2004; Karim and Islam,
2019):

– given a statistical parametric model with density $\mathbf{g}(x, \theta)$ (i.e. an assumed theoretical distribution);

– given a sample $x$ of size $\mathbf{n}$;

to estimate the parameters $\widehat{\theta}$ such that the observations $x$ are the most likely under $\mathbf{g}(x, \widehat{\theta})$.

In the simples 1D case (estimation of the parameter of a one-parameter distribution), the likelihood can be described as a chain
product of probabilities carried out on all n individuals in the sample:





$$L(\theta|x, \mathbf{g}) = \prod_{i=1}^{n} \mathbf{g}(x_i, \widehat{\theta}) \tag{4}$$

Given the fixed (assumed) distribution type $\mathbf{g}$ with parameter $\theta$ and a sample $x$, the likelihood will be the product of all the probabilities $\mathbf{g}(x_i, \widehat{\theta})$, i.e. the distribution's Probability Density Function (PDF) calculated at value $x_i$ ($P(\theta|x_i)$). In practice, since $\mathbf{g}$ and $x$ are fixed, the likelihood is function only of the distribution parameter $\theta$. The maximum of this function will be $\widehat{\theta}$, i.e. the parameter that maximizes the likelihood for that **model type and data combination**. This process can be extended to estimate multiple parameters, increasing the complexity, but the core concept remains unaltered. However, eq. 4 is valid only for complete datasets because the probabilities associated with right-censored data cannot be calculated using the PDF. To solve this problem, it is possible to calculate the MLE for censored datasets, under the assumption of random censoring, using the model's Survival Function (SF) $S(\theta|x_i)$ instead of the PDF for the censored data (Karim and Islam, 2019):

$$L(\theta|x, \mathbf{g}) = \prod_{i=1}^{n} P(\theta|x_i)^{\delta_i} \times S(\theta|x_i)^{1-\delta_i} \tag{5}$$

where $\delta$ is an on-off switch for complete ($\delta = 1$) or censored values ($\delta = 0$). In other words, when a datum is censored, we use the probability information that the individual survived up to the censoring event, given by the survival function.

MLE is a powerful estimator however has its limitations. For example, if the model has more than one parameter, the weight of influence of each parameter are not known and thus it is difficult to know which is the parameter that is controlling the fit. Secondly, simplistic MLE algorithms output a single likelihood value, with a possibility that the optimization gets trapped in a local minimum. This ultimately leads to questioning whether the estimated parameter or combination of parameters is/are the absolute best or if there are other more optimal solutions. These (and other) problems culminate to an even greater uncertainty related to the choice of the distribution, since in theory an infinite variety of functions can be used. These uncertainty tie nicely to the following chapter discussing model selection comparison criteria and how we propose to solve this problem.

### 3.5 Model selection

After fitting the data on a series of models, natural questions such as **"which model fits best?"** or "**which model is the most representative of the data?**" arise. These questions can be reduced to "**having a random sample, is there a way to quantify a distance between a selected model (hypothesis) and the true underlying model?**". In literature, this question is quite often answered with a specific type of null hypothesis tests defined as goodness-of-fit (GoF) test (Storti et al., 2022). However, we would like to point out that these types of tests do not really quantify a distance, but instead define how likely it is that the sample data come from an underlying population **exactly** corresponding to the chosen parametric model. When considering populations of natural objects that are produced by a multitude of processes, this is a fundamentally misleading assumption because it is implicitly assumed that (i) the chosen distribution is the "true underlying model" and thus that (ii) the variability of natural phenomena (many competing processes generating fracture traces in our case) are explained by a





simplified underlying model. Moreover, GoF tests usually have underlying obscure assumptions that undermine their accuracy if not taken into account (Storti et al., 2022). As an alternative to GoF tests, we propose a combined visual and a quantitative approach that can enable the researcher to an informed and guided choice on selecting the most representative model out of a list of *sensible* candidates.

### 3.5.1 A visual approach using the Probability Integral Transform

In statistics the Probability Integral Transform (PIT) is a well-known visual transformation of continuous distributions which states that:

1. given a random sample **X** from a continuous distribution **Y**;

2. given $F_y$ the CDF of **Y**;

3. given $Z = F_y(X)$

**The frequency of Z** is distributed following a standard uniform (Fisher, 1930).

In Fig. 7 a synthetic example of PIT is presented. A random sample **X** of 10000 (non-censored) measures have been extracted from a standard normal distribution **Y** (Fig. 7A), then a standard normal $Y_N$ and exponential $Y_E$ models (hypotheses) have been fit with MLE to the random sample. Figures 7B and 7C represent the empirical frequency histogram and the empirical CDF (ECDF) of **Z** for both models ($Z_N = F_n(X)$ and $Z_E = F_e(X)$). Both figure show that the normal estimated model follows a standard uniform while the exponential model does not. The ECDF visualization of Fig. 7C is preferred over the simple frequency binning of Fig. 7B since multiple models can be tidily represented with the reference uniform distribution appearing as the diagonal of the plot. The closer the ECDF of **Z** is to the diagonal of the plot, the closer the fit is to the true underlying model. The parts of the curve lying below the reference diagonal overestimate the CDF of the true model, and vice versa. Hence the PIT provides a simple visual, yet powerful method to estimate which parametric model better fits the empirical data.

### 3.5.2 A quantitative approach using distances

While the PIT visual approach is very intuitive, we propose to also use a quantitative approach, calculating four different statistics. The first is the Akaike Information Criterion (**AIC**) (Akaike, 1974) that, by using the result of the MLE, quantifies the distance between the true natural phenomena and the estimated model as:

$$AIC = 2k - 2\ln\widehat{\theta} \tag{6}$$

where $k$ is the number of parameters used by the model (i.e. the dimension of the parameter vector) and $\widehat{\theta}$ is the k-dimensional parameter vector that maximizes the likelihood.

This formulation outputs a negative number $(-\infty; 0]$ that approaches zero as the distance between the true population and the





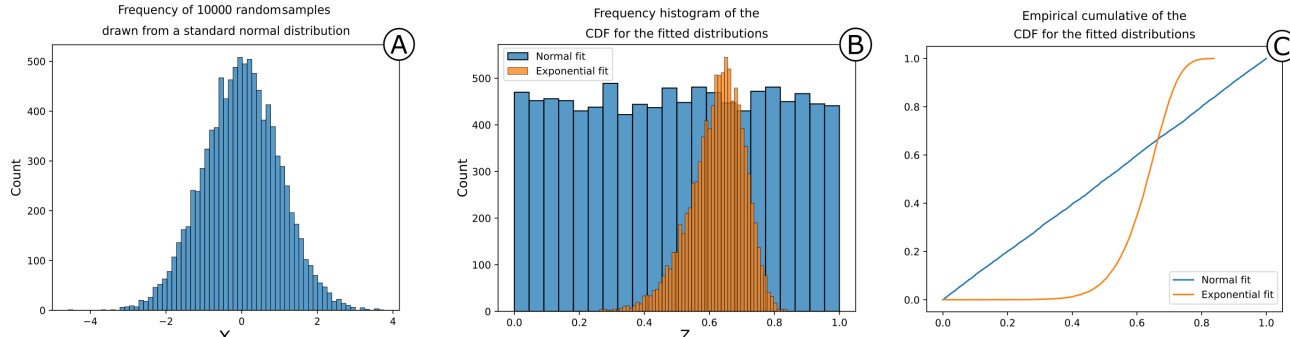

**Figure 7.** Synthetic example of the Probability Integral Transform. In (A) 10000 random samples **X** have been drawn from a standard normal distribution **Y**. In (B) the frequency distribution of **Z** from two different models (hypotheses): normal ($Y_N$) and exponential ($Y_E$). In (C) the empirical cumulative of **Z** for both models. From figure B and C it is possible to observe the effect of the Probability Integral Transform. In B the empirical frequency histogram of the normal model is remarkably close to a uniform distribution while the exponential model is not. This is visualized much more clearly in C where the normal model is the closest to the diagonal line $y = x$ (i.e. the standard uniform).

model decreases. If multiple models are tested, then it is possible and advised to calculate the $\Delta_i$ parameter i.e. the distances between the different models to the best scoring one ($AIC_{min}$):

$$\Delta_i = AIC_i - AIC_{min} \tag{7}$$

where $i$ is the index of the proposed model.

From $\Delta_i$ it is possible to obtain the weight of evidence ($w_i$) of the given model with the following formula:

$$w_i = \frac{e^{-\Delta_i/2}}{\sum_{r=1}^{R} e^{-\Delta_R/2}} \tag{8}$$

where R is the total number of tested models (i.e. models deemed as reasonable by the researcher).

The weight of evidence outputs a value between $[0; 1]$ that represents, in a set of proposed hypotheses, how likely it is that the model comes close to the true underlying process. The closer to 1 the more likely it is that, in the pool of candidates, the model represents the true underlying process. Since AIC and the derived formulas are directly based on the MLE, they are affected by

the same limitations discussed in chapter 3.4 (Akaike, 1974). We thus propose to calculate three different distances between the model and the data (Kim, 2019). Usually, distances are calculated by comparing the empirical cumulative frequency with the cumulative frequency of the model; however, the data are censored and thus we use the empirical cumulative frequency estimated from Kaplan-Meier. Moreover, Kim (2019) proposes to calculate the distances using the data transformed with PIT (i.e. $Z = F_y(X)$). We would like to highlight that purely under the point of view of the calculation, the distances are the same

with or without using the transform, however with PIT the data are "normalized" and thus the different models are compared over the same scale (0, 1). Formula 3 then can be rewritten as:





$$\widehat{G}_n(z) = \begin{cases} 0, & z < Z_1 \\ 1 - \prod_{Z_r \leq z} (\frac{N-r}{N-r+1})^{\delta_r}, & z \leq Z_N \\ 1, & z > Z_N \end{cases} \tag{9}$$

where $r$ are the indexes of each data point and $\delta_r$ is the same on-off switch of Eq. 5

Kim (2019) then proposes to calculate:

– The Kolmogorov-Smirnov statistic ($DC_n$), representing the maximum distance between the two cumulative curves (Kolmogorov, 1933; Smirnov, 1939);

– The Koziol and Green statistic ($\Psi^2$), representing the sum of squared distances between the two cumulative curves (Koziol and Green, 1976);

– The Anderson-Darling statistic ($AC_n^2$), representing the weighted sum of squared distances between the two cumulative
curves, imparting more weight than the Kolmogorov-Smirnov on tail observation (the closer to 0 or 1 the higher the weight; Anderson and Darling (1954)).

The three statistics can be modified to accommodate the presence of censored data by using the Kaplan-Meier estimator.
The Kolmogorov-Smirnoff statistics, generally calculated as:

$$DC_n = \max(DC_n^+, DC_n^-) \tag{10}$$

where with censored data:

$$DC_n^+ = \max(\widehat{G}_n(Z_r) - Z_r) \tag{10a}$$
$$DC_n^- = \max(Z_{r+1} - \widehat{G}_n(Z_r)) \tag{10b}$$

The Koziol-Green statistics is a generalization of the Cramér-von Mises statistics (Koziol and Green, 1976):

$$\Psi^2 = N \int_0^1 (\widehat{G}_n(z) - z)^2 dz \tag{11}$$

Which can be written as (Appendix 2 in Koziol and Green (1976)):

$$\Psi^2 = \frac{1}{3}N + N \sum_{r=1}^N \widehat{G}_n(Z_r) \times (Z_{r+1}) - Z_r \times [\widehat{G}_n(Z_r) - (Z_{r+1} - Z_r)] \tag{11a}$$





Finally, the Anderson-Darling distance (Anderson and Darling, 1954), generally defined as:

$$AC_n^2 = N \int\limits_0^1 \frac{(\widehat{G}_n(z) - z)^2}{z(1-z)} dz \qquad (12)$$

where $\frac{1}{z(1-z)}$ is the weight, can be calculated accounting for censoring as a finite sum (Kim, 2019):

$$AC_n^2 = N \sum_{r=1}^{N-1} (\widehat{G}_n(Z_r))^2 \times (-\ln(1-Z_{r+1}) + \ln(Z_{r+1}) + N \times (1-Z_r) - \ln(Z_r)) +$$

$$-2N \sum_{r=1}^{N-1} (\widehat{G}_n(Z_r)) \times (-\ln(1-Z_{r+1}) + \ln(1-Z_r)) +$$

$$-N \times \ln(1-Z_N) + N \times \ln(Z_N) + N \qquad (12a)$$

Once these distances are calculated for each model, we propose to rank the models independently for each distance (from minimum to maximum). If all distances converge (i.e. for the same model the same rank is assigned) then we consider this as sufficient proof for the overall ranking of the model in the list. If different distances rank in different positions, then by comparing multiple rankings we can still make a sensible guided choice, for example by using the PIT representation together 295 with the mean ranking position or a specific type of distance.

## 4 Case studies

We demonstrate the applicability of all the discussed theory by using the FracAbility (https://github.com/gecos-lab/FracAbility) package on three different case studies. The first one focuses on the characterization of a very regular and densely fractured fracture network with a simple boundary geometry. The second focuses on analysing a more typical outcrop, with multiple 300 fracture sets and complex boundary geometry. Finally, the third case study presents an application with spacing analysis to demonstrate that any "type" of length can (and must) be corrected from right-censoring bias. For the real data applications we propose to test six statistical models of length, frequently adopted in literature (Bonnet et al., 2001):

- Lognormal

- Truncated power law (always defined as *power law* in the rest of the work)

- Normal

- Weibull

- Exponential

- 2-parameters Gamma (always defined as *gamma* in the rest of the work)



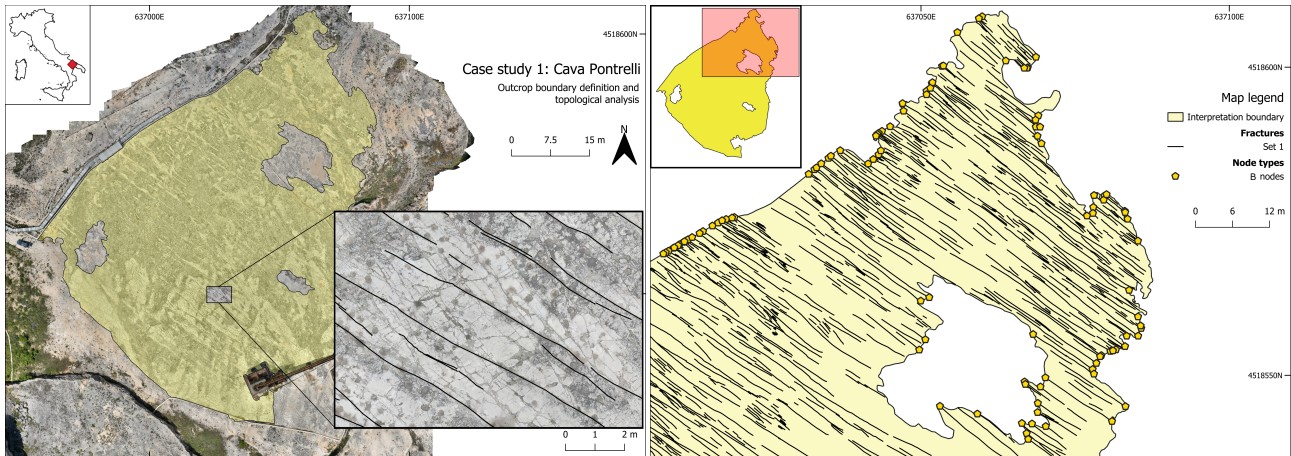

**Figure 8.** Overview maps of the first case study area. Pictured on the left, the general overview of the area with the boundary geometry overlayed on the orthophoto and a small sample of the digitized facture set. On the right it is represented a subarea of the quarry with the boundary, fracture traces and intersection nodes between the two (yellow pentagons).

## 4.1 First case study: Cava Pontrelli, Puglia (IT)

This first case study focusses on the analysis of a single NW-SE striking fracture set present in an abandoned quarry near the town of Altamura in Puglia (Italy). The study area is located in the Apulian Platform, representing the forebulge of the southern part of the Apennines fold and thrust belt (Vai and Martini, 2001; Patacca et al., 2007). The outcrop is characterized by an extensive horizontal pavement of about 18.000m$^2$, showing densely fractured Cretaceous platform limestone of the Altamura Limestone Formation (Ricchetti and Luperto Sinni, 1979). The continuous maintenance that followed the discovery

of thousands of dinosaur footprints on the quarry pavement (Nicosia et al., 1999), made it possible to obtain an exceptionally clean outcrop surface. This in turn resulted in the definition of a simple boundary geometry (with a couple of interpretational holes) and the digitalization of a very regular and dense fracture network (1941 fractures). The combination of these factors led to a very low percentage of censoring with only 8.9% of the total fractures being censored (Fig. 8).

  We applied survival analysis as implemented in FracAbility, and in Fig. 9 the resulting PIT plot is shown. From this, the most

representative model appears to be the lognormal, confirmed by all the distances calculated in Table 1 (ordered by ascending Akaike scores) (see Fig. 10 for a summary of the fit). Figure 9 shows that the lognormal model is quite linear with gentle undulations indicating a slight underestimation for lengths between 1m and 2m (accounting for 20% of the measures) and slight overestimation for 3.5m and 6.5m (accounting for another 20% of the measures). For the calculated distances scores, the lognormal is followed by the gamma and Weibull models, however the rank scores of the different positions do not converge.

While the Akaike distance rank the gamma model at the second position, the other distances do not state the same. The Koziol and Green distance indicates a smaller overall distance between the model and the empirical data for the gamma, but the



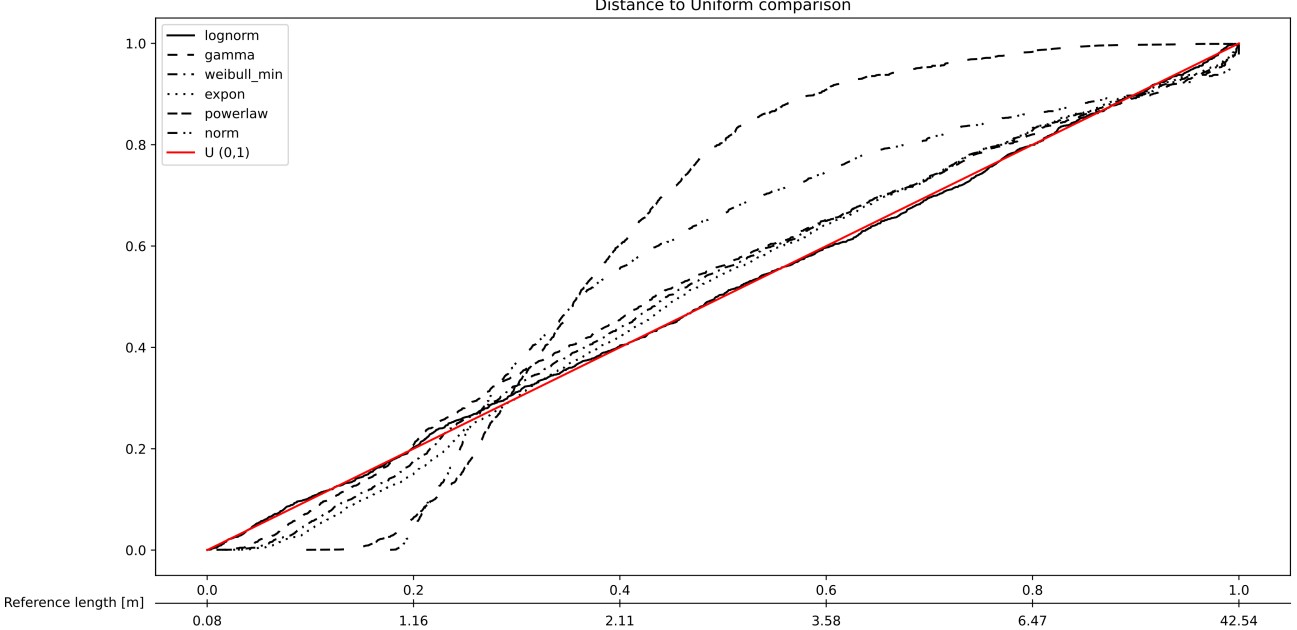

**Figure 9.** PIT visualization for the proposed length models in the Pontrelli quarry. In red represented the reference U(0,1), the closer the model line to this reference line the more representative the model. For this dataset, the lognormal model is the most representative following almost perfectly the reference line with some minor underestimation between 1.16m and 2.11m and overestimation between 3.58m and 6.47m.

**Table 1.** Models' distance tables and ranking scores tables for the Pontrelli quarry. The closer to 0 the better. For this dataset the lognormal is the most representative of the data in all the different distances while the powerlaw and normal models are the least representative. The positions of the other models in between are less certain (especially the gamma and exponential).

| Model | AIC | $\Delta_i$ | $w_i$ | $DC_n$ | $\Psi^2$ | $AC^2$ | AIC rank | $DC_n$ rank | $\Psi_n^2$ rank | $AC_n^2$ rank | Mean rank |
|---|---|---|---|---|---|---|---|---|---|---|---|
| **Lognormal** | 8522.15 | 0.00 | 1.00 | 0.02 | 0.07 | 0.56 | 1 | 1 | 1 | 1 | 1 |
| **Gamma** | 8742.20 | 220.05 | 0.00 | 0.07 | 2.97 | 19.53 | 2 | 4 | 4 | 3 | 3 |
| **Weibull** | 8770.37 | 248.22 | 0.00 | 0.06 | 2.53 | 19.18 | 3 | 2 | 3 | 2 | 3 |
| **Exponential** | 8774.85 | 252.70 | 0.00 | 0.06 | 2.33 | 19.53 | 4 | 3 | 2 | 4 | 3 |
| **Power law** | 10639.54 | 2117.40 | 0.00 | 0.32 | 71.71 | 354.20 | 5 | 6 | 6 | 6 | 6 |
| **Normal** | 10682.11 | 2159.97 | 0.00 | 0.18 | 24.54 | 148.01 | 6 | 5 | 5 | 5 | 5 |

Weibull seem to have a smaller distance considering the tails as in the Anderson-Darling distance. Finally, the power law and normal models always occupy the last two positions.



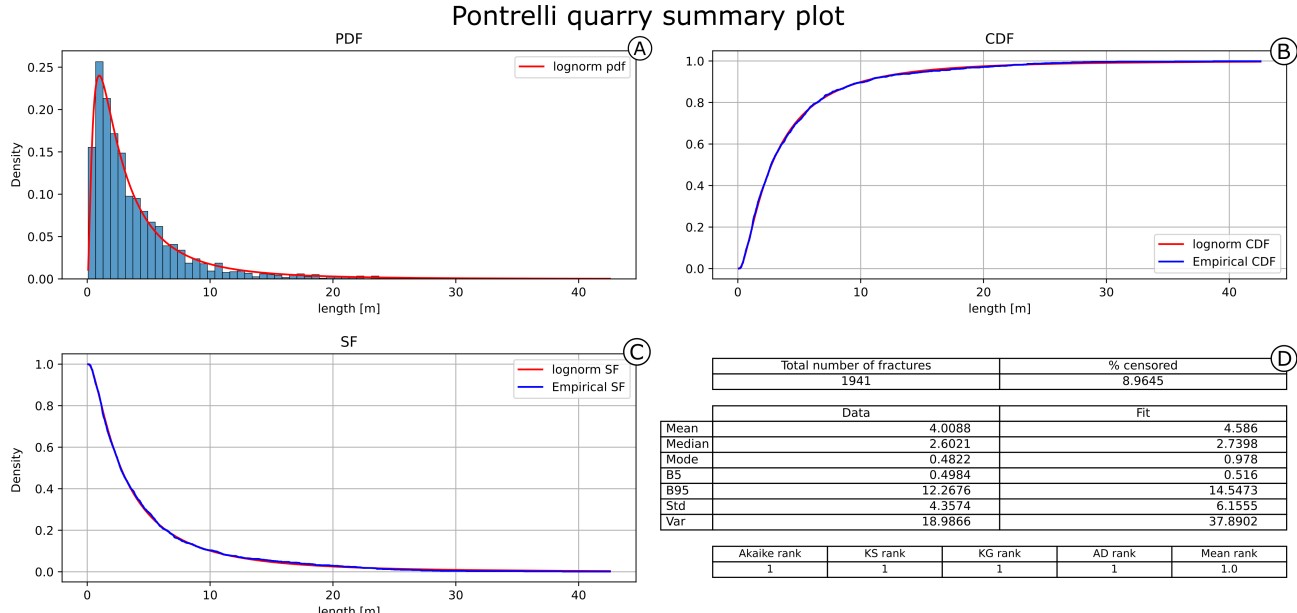

**Figure 10.** Summary of the best fitting model (lognormal) for the Pontrelli quarry dataset. (A) The probability density function against the histogram of the dataset. (B) Cumulative Density Function and the (C) Survival Function against the empirical counterparts calculated with the Kaplan-Meier estimator. (D) The summary table of the main statistics for the data (e.g. sample mean, sample variance etc.) compared with the estimated model.



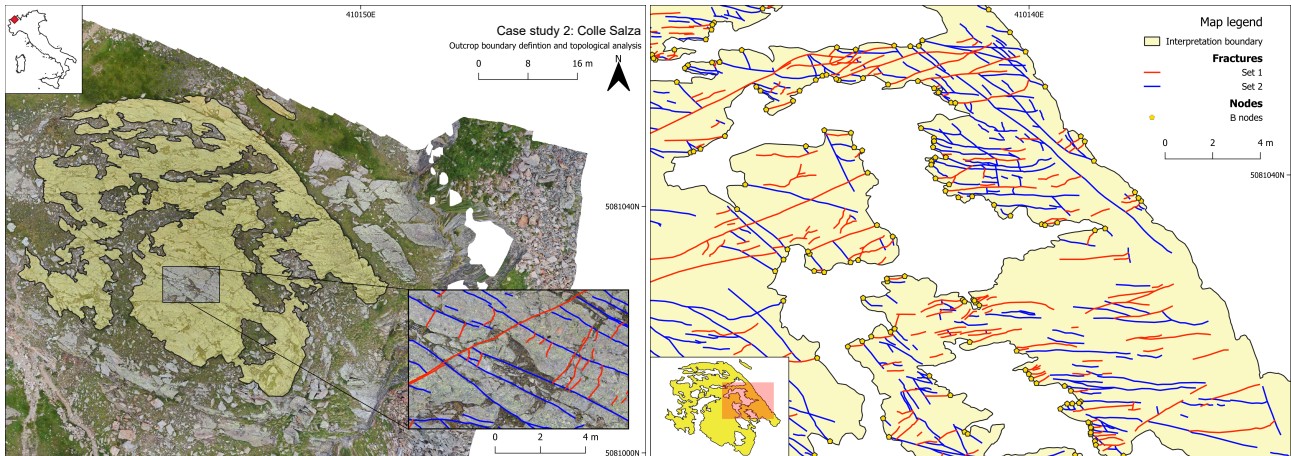

**Figure 11.** Overview maps of the second case study area. Pictured on the left, the general overview of the area with the boundary geometry overlayed on the orthophoto and a small sample of the digitized facture sets. On the right it is represented a subarea of the quarry with the boundary, fracture traces and intersection nodes between the two (yellow pentagons).

## 4.2 Second case study: Colle Salza, Valle d'Aosta (IT)

The second case study focuses on the analysis of a less ideal, albeit more realistic ourcrop. The study area is located in the basement of the Western Alps (Dal Piaz et al., 2003), on paragneiss of the Monte Rosa Nappe (Dal Piaz and Lombardo, 1986). The outcrop is cross-cut by several brittle fractures, Tertiary in age (Bistacchi and Massironi, 2000), and is characterized by a main central area of $1234m^2$ and two secondary smaller satellite areas of $9m^2$ and $3m^2$. The boundary geometry is highly convex thus leading to a high censoring fraction of fracture traces. Moreover, the outcrop is exposed on top of a small

topographical height with a slightly ellipsoidal shape due to glacial erosion (i.e. a *roche moutonnée*) with the main axis directed NW-SE. This led to an inevitable deformation of the orthophoto (and thus of the measured lengths) along the extremities of the analyzed area. The analysed fractures were subdivided in two main sets, striking NE-SW (Set 1) and NW-SE (Set 2), conforming to the general trend of the area for brittle deformation (Bistacchi et al., 2000; Bistacchi and Massironi, 2000; Bistacchi et al., 2001). The total number of fractures is 1718 (24.7% censored) while set-wise the number of fractures is 692

(24.3% censored) for set 1 and 1026 (24.9% censored) for set 2 (Fig. 11).

In Fig. 12 the different estimated models are represented for both sets. The PIT visualization shows how in both cases the lognormal is the model that, overall, better fits the data by being generally the closest to the reference line. Nonetheless, the fit of estimated models is visibly worse than in the first case study, particularly for set 1 (Fig. 12A). The lognormal fit of set 1 underestimates lengths between 0.34m and 1.5m (about 50% of the measures) while lengths above this value are generally

overestimated. For the lognormal fit of set 2 (Fig. 12B), there is a less relevant underestimation of the length values between 0.44m and 2.57m (about 60% of the measures) and a slight overestimation afterwards. The distances values in Table 2 (ordered by ascending values of Akaike) confirm the lognormal as the most representative for both sets (see Fig. 13 for the summary





**Table 2.** Models' distance and rank tables for both Colle Salza sets. The closer to 0 the better. For this length dataset, the lognormal is the most representative of the data for all the different distances while the gamma model is ranked lower for the $DC_n$, $\Psi_n^2$ and $AC_n^2$ ranks, indicating a worst fit in respect of the exponential and Weibull distributions positioned in second and third place respectively.

| | | | | | | | | | | | Set 1 |
|---|---|---|---|---|---|---|---|---|---|---|---|
| Model | AIC | $\Delta_i$ | $w_i$ | $DC_n$ | $\Psi^2$ | $AC^2$ | AIC rank | $DC_n$ rank | $\Psi_n^2$ rank | $AC_n^2$ rank | Mean rank |
| **Lognormal** | 1118.98 | 0.00 | 1.00 | 0.06 | 0.55 | 5.05 | 1 | 1 | 1 | 1 | 1 |
| **Gamma** | 1264.59 | 145.61 | 0.00 | 0.12 | 2.67 | 23.59 | 2 | 4 | 4 | 4 | 3.5 |
| **Exponential** | 1280.38 | 161.40 | 0.00 | 0.10 | 2.32 | 20.82 | 3 | 2 | 2 | 2 | 2.25 |
| **Weibull** | 1281.01 | 162.03 | 0.00 | 0.11 | 2.40 | 21.63 | 4 | 3 | 3 | 3 | 3.25 |
| **Power law** | 1935.96 | 816.98 | 0.00 | 0.36 | 32.68 | 160.01 | 5 | 6 | 6 | 6 | 5.75 |
| **Normal** | 2068.24 | 949.26 | 0.00 | 0.22 | 11.84 | 79.92 | 6 | 5 | 5 | 5 | 5.25 |

| | | | | | | | | | | | Set 2 |
|---|---|---|---|---|---|---|---|---|---|---|---|
| Model | AIC | $\Delta_i$ | $w_i$ | $DC_n$ | $\Psi^2$ | $AC^2$ | AIC rank | $DC_n$ rank | $\Psi_n^2$ rank | $AC_n^2$ rank | Mean rank |
| **Lognormal** | 2204.06 | 0.00 | 1.00 | 0.04 | 0.39 | 4.53 | 1 | 1 | 1 | 1 | 1 |
| **Gamma** | 2379.13 | 175.07 | 0.00 | 0.09 | 2.97 | 29.94 | 2 | 4 | 4 | 4 | 3.5 |
| **Exponential** | 2390.81 | 186.75 | 0.00 | 0.09 | 2.54 | 25.31 | 3 | 2 | 2 | 2 | 2.25 |
| **Weibull** | 2392.34 | 188.28 | 0.00 | 0.09 | 2.62 | 26.24 | 4 | 3 | 3 | 3 | 3.25 |
| **Power law** | 3109.95 | 905.89 | 0.00 | 0.31 | 35.29 | 167.56 | 5 | 6 | 6 | 6 | 5.75 |
| **Normal** | 3421.72 | 1217.66 | 0.00 | 0.21 | 15.24 | 137.05 | 6 | 5 | 5 | 5 | 5.25 |

plots of the best fit). For the other models, looking at the mean rank value helps in understanding the final ranking showing that the gamma distribution is ranked lower than the exponential and the Weibull (at the second and third place respectively).





**Figure 12.** PIT visualization for the proposed length models for the set 1 (A) and set 2 (B) of the Colle Salza dataset. In red represented the reference U(0,1), the closer the model line to this reference line the more representative the model. For both sets, the lognormal model is the closest to the reference line although it shows in both sets a worst fit than the first case study. In both sets, all the estimated models are less linear, particularly underestimating between 0.34m and 1.5m in (A) and 0.44m and 2.57m in (B).



## Colle Salza set 1 summary plot

## Colle Salza set 2 summary plot

**Figure 13.** Summary of the best fitting model for the Colle Salza dataset (set 1 and 2). (A) The probability density function against the histogram of the dataset. (B) The Cumulative Density Function and the (C) Survival Function against the empirical counterparts calculated with the Kaplan-Meier estimator. (D) A summary table of the main statistics for the data (e.g. sample mean, sample variance etc.) and estimated model.





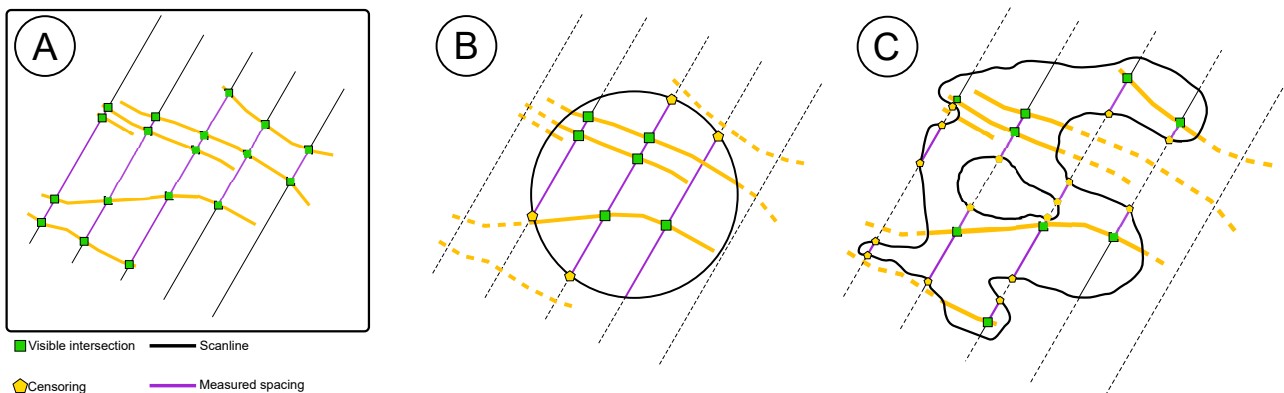

Visible intersection ▬▬▬ Scanline

Censoring ▬▬▬ Measured spacing

**Figure 14.** The effect of boundary geometry on censoring for spacing distribution modelling. (A) The ideal situation where no censoring occurs. Coloured in green are the visible intersections between the scanline and the fracture while in blue the measured spacing segment. (B) Shows the effect of a perfectly convex boundary (i.e. a circle) on the estimation of censoring. For the "visible" scanlines only the final segments are censored. (C) The effect of a complex boundary geometry. Here the number of censoring measurements increases drastically and that censoring it is no longer limited to the ends of the scanlines.

## 4.3  Third case study: Spacing

With the Colle Salza dataset, we also show that survival analysis can be used to analyse the spacing distribution for each fracture set. Spacing is defined as the distance between two fractures of the same set, measured perpendicular to the average fracture plane attitude (Bistacchi et al. (2020) and refs. therein). Traditionally, this type of statistic is obtained in the field with scanline surveys, with properly oriented scanlines or by applying the Terzaghi correction (Terzaghi, 1965). With DOMs, this procedure can be easily sped up with custom scripts, programmatically tracing a large number of parallel scanlines and thus increasing the number of spacing measurements (De Toffoli et al., 2021; Storti et al., 2022). However, this leads to a higher number of censored spacing measurements occurring at the outcrop boundary and where "holes" are present. As with standard lengths, the introduction of a boundary affects the actual estimation of the spacing by a degree depending on the outcrop's convexity (Fig. 14). Perfectly convex boundaries (Fig. 14B) will lead to minor censoring, affecting only the ends of the scanlines, while more realistic non-convex boundaries will increase the censoring percentage because the boundary will cross in multiple points the same scanline (Fig 14C). For the Colle Salza dataset, 1282 and 1119 spacing measurements have been obtained for fracture sets 1 and 2 respectively (with 52.5% and 36.5% censoring). Figure 15 shows the distances of the different models for the obtained spacing values. In this case, for set 1, an extreme underestimation of the lognormal model is present for values longer than 4 m (Fig 15A), while for the other proposed models, particularly the Weibull, the distances at the same values are lower. This is visible also in the results of the distance tables (Table 3). Although the Akaike distance for the lognormal model is the smallest, the difference from the other models ($\Delta_i$) is not as big as the other examples thus the weight of evidence ($w_i$) also is not completely in favor of the lognormal. The bad fit of the lognormal model is also confirmed by the other distances that position the lognormal lower than the Weibull and gamma models (at first and second place respectively). All these factors





**Table 3.** Models' distance and rank tables for the spacing calculated on both Colle Salza sets. The closer to 0 the better. For set 1 the low performance of the lognormal in the estimation of longer values (Fig. 15A) is visible in all distances. Even if the Akaike distance calculated for the lognormal is the smallest, the difference from the other calculated values is not enough to completely justify the selection of this model. This is also shown in the other distances where the lognormal clearly underperforms. On the other hand, for set 2 the lognormal model is the most representative for the spacing values for all the different distances.

| | | | | | | | | | | | |
|---|---|---|---|---|---|---|---|---|---|---|---|
| | | | | | Set 1 | | | | | | |
| Model | AIC | $\Delta_i$ | $w_i$ | $DC_n$ | $\Psi^2$ | $AC^2$ | AIC rank | $DC_n$ rank | $\Psi_n^2$ rank | $AC_n^2$ rank | Mean rank |
| **Lognormal** | 2508.66 | 0.00 | 0.55 | 0.11 | 1.25 | 13.32 | 1 | 4 | 3 | 4 | 3 |
| **Weibull** | 2509.59 | 0.93 | 0.35 | 0.05 | 0.75 | 6.17 | 2 | 1 | 1 | 1 | 1.25 |
| **Gamma** | 2512.84 | 4.17 | 0.07 | 0.06 | 1.03 | 8.37 | 3 | 2 | 2 | 2 | 2.25 |
| **Exponential** | 2514.14 | 5.48 | 0.04 | 0.07 | 1.65 | 13.27 | 4 | 3 | 4 | 3 | 3.5 |
| **Power law** | 2682.15 | 173.49 | 0.00 | 0.16 | 12.65 | 116.42 | 5 | 6 | 6 | 5 | 5.5 |
| **Normal** | 3247.15 | 738.49 | 0.00 | 0.14 | 10.45 | 169.38 | 6 | 5 | 5 | 6 | 5.5 |
| | | | | | Set 2 | | | | | | |
| Model | AIC | $\Delta_i$ | $w_i$ | $DC_n$ | $\Psi^2$ | $AC^2$ | AIC rank | $DC_n$ rank | $\Psi_n^2$ rank | $AC_n^2$ rank | Mean rank |
| **Lognormal** | 1652.30 | 0.00 | 1.00 | 0.03 | 0.15 | 1.47 | 1 | 1 | 1 | 1 | 1 |
| **Gamma** | 1709.71 | 57.41 | 0.00 | 0.07 | 1.40 | 23.53 | 2 | 4 | 4 | 4 | 3.5 |
| **Exponential** | 1712.78 | 60.48 | 0.00 | 0.06 | 1.18 | 19.66 | 3 | 2 | 2 | 2 | 2.25 |
| **Weibull** | 1714.62 | 62.31 | 0.00 | 0.06 | 1.22 | 20.53 | 4 | 3 | 3 | 3 | 3.25 |
| **Power law** | 2340.23 | 687.93 | 0.00 | 0.31 | 38.10 | 180.73 | 5 | 6 | 6 | 6 | 5.75 |
| **Normal** | 2603.07 | 950.77 | 0.00 | 0.17 | 11.56 | 136.26 | 6 | 5 | 5 | 5 | 5.25 |

point to the fact that in this case the Weibull is the most representative model (see Fig. 16 for a summary representation of the best fit). On the other hand, for set 2 the lognormal is quite clearly the most representative (Fig 15B) with the PIT plot showing a quite linear behavior and the distances ranking converging all at the first place.



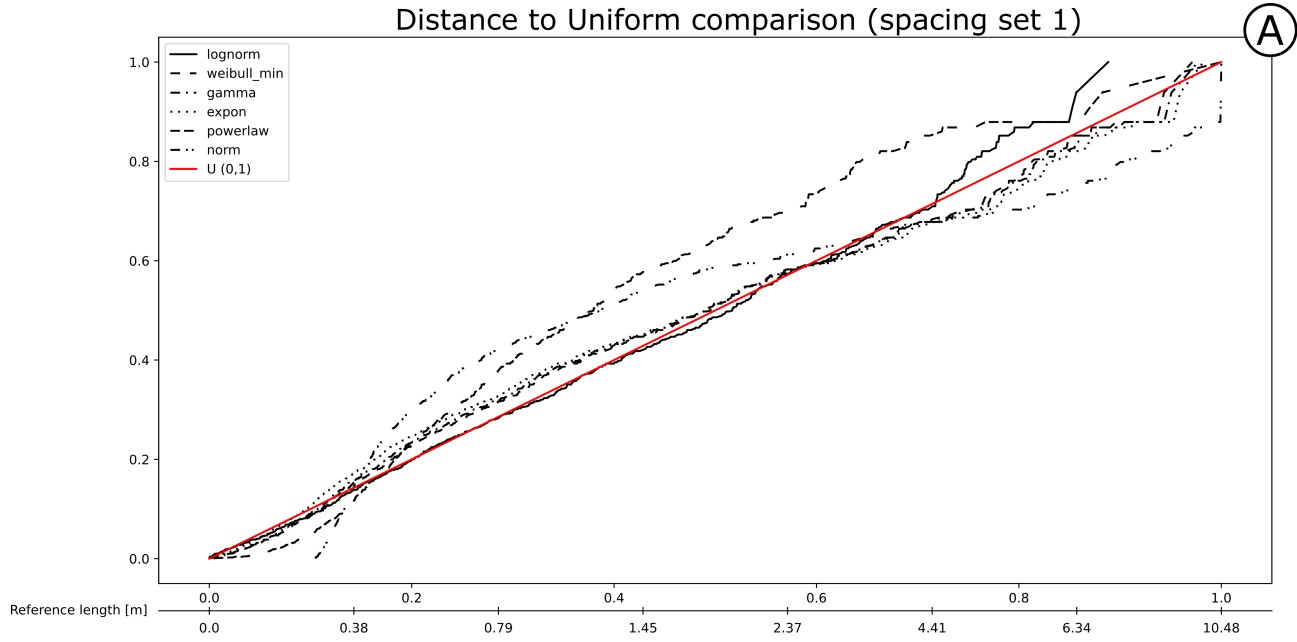

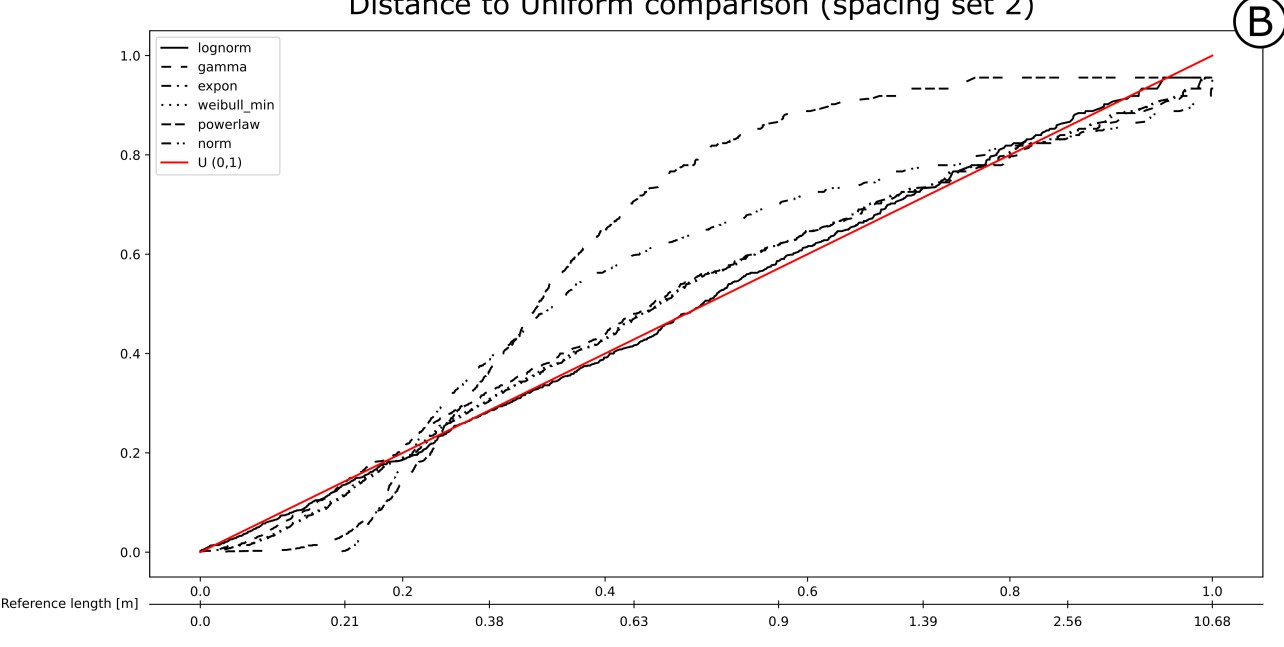

**Figure 15.** PIT visualization for the proposed length models for the spacing values calculated on set 1 (A) and set 2 (B). In red represented the reference U(0,1), the closer the model line to this reference line the more representative the model. In (A) the lognormal model presents a marked underestimation for values longer than 4.4m. In (B) on the other hand the lognormal model is the closest to the reference line performing quite well for the spacing measurements.





**Figure 16.** Summary of the best fitting model for the spacing dataset (set 1 and 2). (A) The probability density function against the histogram of the dataset. (B) The Cumulative Density Function and the (C) Survival Function against the empirical counterparts calculated with the Kaplan-Meier estimator. (D) A summary table of the main statistics for the data (e.g. sample mean, sample variance etc.) and estimated model. The spacing of set 1 is the only case where the data is not lognormally distributed and the frequency histogram reflects such conclusion.



## 5   Discussion

With this work we delineate a robust statistical framework to quantitively analyse and statistically model fracture trace length (and other parameters such as spacing). The statistical distributions are estimated using the entire length of fracture traces, following the original works of Baecher and Lanney (1978); Baecher (1980) and not the length of branches as defined in Sanderson and Nixon 2015. Furthermore we consider censoring bias and the possibility to treat the effect of censoring on the estimation of fracture trace length by applying survival analysis.

A crucial point in discussion is the objective of having a representative unbiased statistical model that is most likely to reproduce the observed length data. In particular, we want to stress that there is **no real interest, nor practical reason**, and **no theoretical possibility** to find the **real underlying distribution**, while there is a **strong necessity to fit specific simplified parametric models**, useful for stochastic modelling applications down the line (such as DFNs). Because of this reason, non-parametrical methods such as those proposed by Mauldon et al. (2001) and implemented in software such as FracPaQ (Healy et al., 2017) are unfit since they are not linked to any model. Furthermore, having an unbiased statical model is also extremely useful for engineering approaches. For example, Pahl (1981) states at the end of his work that in geomechanical application it could be useful to know the longest trace likely to occur in a group of traces. With a parametric model this can be easily estimated by checking the length values associated to a probability chosen depending on the safety margin that is needed for the use case. Approaches such as ignoring censoring or removing censored data do provide a statistical distribution, however the censoring bias is still present and thus the results are skewed, always underestimating length. The proposed approach provides for the middle ground that was missing. We have shown that it is possible to use survival analysis to model censored length datasets and we have showcased a new approach in model selection via quantitative distances to rank and choose the most representative model from a pool of simple candidates that can be used in common modelling applications.

As stated, to use MLE for a given model we assumed that, in geological applications, the only censoring mechanism present is random censoring. Both left and interval censoring, by definition, cannot happen because the event of interest coincides with the end of the fracture. Thus, for left censoring, fractures terminate outside the boundary and thus cannot be measured (this is analogous to a patient never entering the study because it is already dead). The same applies with interval censoring that could be erroneously identified when holes are present in the sampling area or when boundaries are concave. However, since the measured event is the end of the fracture, on the other side of the interruption no fractures should be present. In terms of classic survival analysis theory, this would compare to a patient's loss of follow-up in the study period, caused by voluntary or involuntary exit from the study group, thus classifying it again as right censoring (Leung et al., 1997).

To correctly classify censoring as random, we must assume independence of the censoring lengths from fractures lengths. This final assumption is difficult, if not impossible, to rigorously prove since we do not know the true distribution of the length of fractures (we only observe a set of complete and censored data). However, it can be safely assumed in geological applications that these two lengths are independent, because the boundary, which causes censoring, is usually the product of secondary events that occur after fracture genesis (i.e. alteration, debris hiding part of the outcrop, vegetation, human activity, etc.) and thus the physical processes causing censoring do not influence the distribution of length which is determined by other, com-

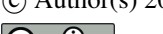



**Table 4.** Summary table comparing mean ($\mu$) and standard deviation ($\sigma^2$) values for increasing censoring fractions. Both the ignore censoring and remove censored data methods underestimate the values with the latter always showing a stronger underestimation.

| Method | Pontrelli (8.96%) | | Salza S1 (24.28%) | | Spacing S2 (36.46%) | | Spacing S1 (52.50%) | |
|---|---|---|---|---|---|---|---|---|
| | $\mu$ | $\sigma^2$ | $\mu$ | $\sigma^2$ | $\mu$ | $\sigma^2$ | $\mu$ | $\sigma^2$ |
| Ignore censoring | 4.077 | 5.032 | 0.9 | 0.895 | 0.844 | 1.225 | 1.373 | 1.453 |
| Remove censored | 3.856 | 4.721 | 0.811 | 0.785 | 0.744 | 0.974 | 1.291 | 1.351 |
| Survival | 4.586 | 6.156 | 1.218 | 1.482 | 1.551 | 2.804 | 3.072 | 3.324 |

pletely independent processes.

As a second point, we would like to discuss is the relationship between the fraction of censored data and the uncertainty in the estimated statistical model. There is not much literature on this relationship probably because, since traditional survival analysis is applied in key of time, the time boundary can be expanded if necessary. For example, if a study shows a censoring rate that is deemed too high, the researcher could repeat the experiment and simply extend the study period to observe the event of interest. In our case, the spatial boundary usually cannot be expanded (if we could, our work would be useless) and thus knowing the effects of censoring on estimation is an important aspect. For example, it would be useful to know how censoring influences the estimations based on simplistic approaches delineated in section 3, or the censoring percentage value above which survival analysis must be used, or the value above which even survival analysis fails. Moreover, it can be also useful to estimate the precision of estimation with survival analysis, knowing the censoring percentage. The case studies discussed in this work show a censoring percentage ranging from 8.96% (first case) to 52.50% (third case), and effectively showcase the effects of censoring on the estimation. For each dataset it is possible to use the best fitting statistical model defined in the results and use PIT to visualize the differences. The comparison with circular line approaches was intentionally omitted because it cannot be compared in any way. Being a non-parametrical approach, cumulative frequencies cannot be compared and comparing mean values without a variance is completely useless. Figure 17 shows that in all cases, the estimated model using survival analysis closely follows the reference uniform from PIT, while the other two approaches (i.e. ignoring censoring and removing censored data) always overestimate the cumulative frequency, diverging with a rate positively correlated with the censored fraction. Moreover, the model obtained by removing censored values is always the worst. This is because MLE is a consistent estimator that converges as the number of data increase, and thus by removing data we are effectively hindering the estimator's precision (in the last case study we would be removing 52% of the data i.e. 667 fractures). This effect is also visible when comparing the estimation of important distribution parameters such as mean ($\mu$) and standard deviation ($\sigma^2$). Table 4 compares the values of these parameters calculated with the different methods, showing a consistent underestimation of both the mean and standard deviation, with a stronger underestimation coming from the removing censored method.

The outlined effect could be useful to predict what to expect from any survival analysis and shed some light on the exploration of the censoring mechanics that govern these types of datasets. Still, the comparison briefly discussed in this chapter is not



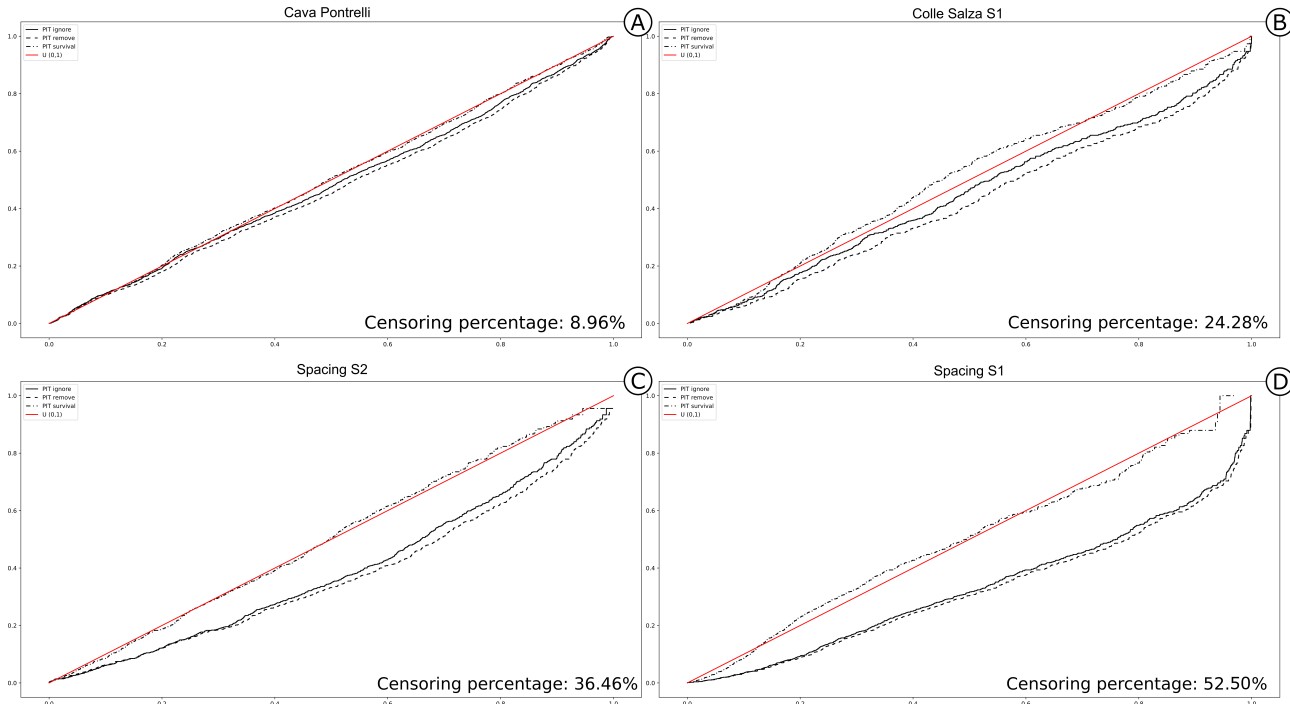

**Figure 17.** PIT visualization for the lognormal model estimated with three different approaches: Ignoring censoring (full line), removing censored data (dashed line), and using a survival analysis approach (dash and dot line). The survival model follows almost constantly the true model (red line) while the other two approaches increasingly diverge on longer fractures with a rate depending on the censored fraction.

rigorous enough to fully constrain this relationship. Even if we have a clear increase in the censoring percentage in the different datasets, the genetic factors, lithology and censoring mechanics are completely different. Moreover, the number of measures in each case study are different and thus not statistically comparable. Because of these reasons, this behaviour should be isolated and modelled in controlled synthetic experiments to be carried out in the future. Nevertheless, we found that our preliminary

results and the possible implications were too interesting not to be discussed.

As a third point, the readers that support and firmly believe in power law distribution of length may be shocked by the results of this paper. In all the different case studies, the power law distribution always ranked last or second to last together with the normal distribution. Power law distributions theoretically describe many natural events, and in our case if fractures for example are self-similar then the lengths could show a power law distribution (Bonnet et al., 2001). However, it is shown that in general

**truncated** power law models better fit natural and realistic data and thus usually only the tails of the distribution follow a power law (Clauset et al., 2009). This entails that to properly fit a power law it is necessary to estimate both the scaling parameter ($\alpha$) **and** the minimum truncation value ($x_{min}$). Truncation is a constantly present bias caused by the resolution limits of the data source above or under which no data can be acquired (right and left truncation respectively). In our case, left truncation (i.e. $x_{min}$) is the most common type and it is caused by the pixel resolution of the images under which fractures traces cannot



be seen. Because of this limitation the modelled length distribution will always have an underestimation of the frequencies for
the shorter lengths and thus usually only the tail of the distribution follows a power law. Adding to this bias is also the human
error caused by countless different factors (one of which is the inevitable boredom of digitizing an outcrop). If the length
distribution truly follows truncated power laws, then shorter fractures near the truncation limit would be much more frequent
however harder to spot. This usually results in fractures that are left uninterpreted even if visible and thus leading to an increase

of the real $x_{min}$ value. Truncation bias and fitting power laws is a very active field of research still with no definitive solution
(Clauset et al., 2009; Deluca and Corral, 2013), thus, the implementation, testing and discussion of such solutions would lie
outside this work. For the same reason a proper estimation procedure could not be included, and the results obtained in this
work relative to the power law model must be taken with a grain of salt. Nonetheless, we still wanted to leave the estimation
results and discussion for power laws to again show, as Healy et al. (2017) did, how this model cannot be blindly applied to

geological data without carrying out necessary important considerations.
Finally, censoring is one of many biases that influence the length measurements, and the presented work only treats this specific
bias. Because of this, if other biases are present, they will be carried out also after the correction. Moreover, the underlying sta-
tistical model between different sets can be different and each fracture set has its own set of biases that influence measurements
and variation in the statistics can also occur within the same set for example in proximity of a fault or local changes in rhe-

ology. Because of all these reasons, grouping all entities in the same statistical model without any kind of consideration leads
to inevitable misinterpretation of the statistics and provide a wrong statistical parametrization of the whole network and thus
the analysis of a fractured rock system must be carried out in an homogeneous domain, or different stationary domains must
be defined before any further analysis is carried out (Bistacchi et al., 2020). By applying this divide and conquer approach, a
more precise and robust characterization is assured because statistical models and inevitable biases related to different families

will not mix. These observations further highlight the crucial importance of field data acquisition and geological reasoning to
avoid blindly applying these methods to outcrops and make severe inferential errors.

## 6    Conclusions

In this paper the effects of censoring bias on the estimation of statistical length models are delineated and discussed. In
particular we demonstrated that censoring bias leads to a general underestimation of any distribution. The typical approaches

found in literature are not suitable and we have shown that using survival analysis is the proper way to treat censoring also
in length datasets. In particular, survival analysis offers a valid alternative to the popular circular scan line method. Firstly,
survival analysis methods are based on the relationship between the event of interest and the censoring mechanism. Without
any underlying geometrical assumptions (basis of the circular scanline method) this methodology is quite flexible and can be
easily applied in any geological case study. Secondly, the parameters that are estimated with survival analysis are directly based

on the measured length (i.e. the quantity that we want to model) and are always associated with a model thus have a higher
statistical significance of the scan line method. Regarding the other parametrical methods (i.e. ignoring censoring and removing
censored data), we have shown that the censoring percentage heavily influences the estimation quality. In particular, with only



8.96% of censoring the two classical methods underestimate the mean and variance and the increase in censoring percentage positively correlates with such underestimation. Also survival analysis is visibly impacted by the increase of censoring, however its output remains always stable around the natural model (i.e. the reference diagonal). Nonetheless, the influence of censoring percentage is yet not fully constrained since in this study the visible variation is also probably given by different datasets, with different genetic factors. Thus, a more robust and in-depth analysis must be carried out to fully understand this relationship. The proposed combination of the PIT visualization and distances calculations demonstrate an effective approach to quantitatively rank a list of length distribution models. The workflow has the objective of comparing sensible (simple) models useful to practical applications (such as DFNs) and find which one best represents the data. The theory and necessary tools to carry out a complete statistical characterization of a fracture network are included in the specifically developed FracAbility Python library (https://github.com/gecos-lab/FracAbility) that encapsulates the necessary steps to carry out as easily and quickly as possible the workflow starting from input shapefiles. The methodology and library were successfully tested in three different use cases. The first having almost no censoring bias and the last with more than 50% of the total measurements being censored. In all cases, except for the last in which one of the two spacing distributions follows a Weibull distribution, the lognormal model was the most accurate followed by either gamma, exponential or Weibull.

*Code and data availability.*   The code and data are available at the GitHub repository (https://github.com/gecos-lab/FracAbility).

*Author contributions.*   Andrea Bistacchi and Mattia Martinelli first tested and proved the possibility of analyzing and model censored fracture length data. Gabriele Benedetti and Stefano Casiraghi formulated the initial key ideas and goals of the manuscript. Gabriele Benedetti, Stefano Casiraghi, Daniela Bertacchi and Andrea Bistacchi designed the model testing procedure. Gabriele Benedetti wrote all the used code, tested it and improved it in collaboration with Stefano Casiraghi. Stefano Casiraghi provided the used datasets. Andrea Bistacchi and Daniela Bertacchi both heavily contributed to defining the final structure and were fundamental in the revision process. Andrea Bistacchi provided the funding used to gather the data, develop the code and to present and publish the results of this work.

*Competing interests.*   The authors declare that there are no competing interests

*Acknowledgements.*   We thank Christian Albertini, Francesco Bigoni, Fabio La Valle, Sylvain Mayolle, Mattia Martinelli and Silvia Mittempergher for the stimulating discussions carried out during the numerous (and long) project meetings. We also thank Eni S.p.A. for the collaboration in our Joint Research Agreement and for funding Stefano Casiraghi PhD scholarship.



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
