# Peer review of "Unbiased statistical length analysis of linear features: Adapting survival analysis to geological applications"

_EGUsphere, 2024_

## Community Comment (CC1)

This manuscript is an important contribution to a topic of considerable interest within geoscience. Fracture length distributions are important for rock strength and permeability, and thus are of great practical interest. Although length distributions from field studies are widely used inputs for modeling, there has long been uncertainty about how best to measure and analyze the outcrop observations.

There are in my opinion a couple of places in the MS where some clarifications will help improve the impact of the contribution. One of these areas is in contextualizing the work in the Introduction (see comments for lines 81, 106). The other is the at the beginning and in the transitions between the explanation of the survival analysis (see line 129 comment). There are also a couple of minor usage issues; I've highlighted some. The comments are keyed to lines in the text.

81 I suggest making this comment more nuanced and adding a reference: 'joints when empty or veins when filled (refs), although many fractures have hybrid fill attributes: they may be partly filled with inconspicuous mineral deposits that resemble joints, or the degree of fill may depend on fracture width, so that small fractures resemble veins (e.g. Laubach et al., 2019).' After all, many fractures of interest to subsurface applications are strictly speaking neither 'joints' nor 'veins'. In some populations small fractures are fill but wide fractures are open with a thin mineral lining. The old joints versus veins terminology is not helpful, and this is particularly germane for the discussion of length since 'open' fracture length may depend on these width-dependent mineral infills. It's better to call them 'opening-mode fractures or faults' and separately specify the fill state. Laubach, S.E., Lander, R.H., Criscenti, L.J., et al., 2019. The role of chemistry in fracture pattern development and opportunities to advance interpretations of geological materials. Reviews of Geophysics, 57 (3), 1065-1111. doi:10.1029/2019RG000671.

90 The ambiguity of lengths where, as is the common case, fractures are segmented and en echelon, ought to be mentioned. This is a big source of uncertainty in measured lengths (and heights) and there are now ways to deal with this rationally with other node types. See the Forstner paper.

106 Here the potential for flow in the fracture network is assumed to be a function of connectivity, but in the preceding list of fracture types many of the elements many not be conducive of fluid flow, for example some faults with gouge and opening-mode fractures that are sealed. Likewise, if you have a situation where sets are of different ages, early sets may be sealed (or partly sealed) and later ones more open. An example is an outcrop of veins abutted or crossed by later joints. These abutting and crossing relations may impart high connectivity but will have a different impact on flow than a bunch of intersecting open joints. Maybe in 106 say: "If all the fractures are open, a network with prevalence of I

nodes…" This may not be central to the point that you are making in this paper, but it's such a common and misleading logical jump in fracture network studies (and with respect to length) that the clarification is useful. See the discussion in Forstner and Laubach, 2022, J. Struct. Geol.

Also, if the rock itself is porous, even a network that has only I nodes can markedly augment fluid flow because of flow between fractures through the host rock (Philip et al., 2005, SPE Res. Eval. Eng.); here length distribution is the key parameter (not connectivity) as Philip et al. show, which just makes your focus on length even more important.

122 It seems like these values might also be meaningless for 'stochastic modeling'? Do you clarify this in the Discussion?

129-132 On first read, I found the transitions here confusing. For clarity I think you ought to warn the reader here that you are going to demonstrate the time-length dimensional shift in 3.3. Something like 'Survival analysis is usually used in the time domain. In section 3.3. we show how a time-length dimensional shift is valid. Here we briefly introduce the terms as they are used in the time domain.' These are key lines defining terms. I think they could use some clarification. What do you mean by 'the event of interest is commonly defined as *death*'? Is a clarifying word missing? The 'event of "x" is'? Or do you need some more information at the start of the paragraph: "Survival analysis is used to analyze data in which the time until the event is of interest (for example, the time until death in some medical or biological contexts)." This would perhaps be a good point to introduce the idea that you are substituting distance for time?

133 Which 'length' do you mean here?

173 1D or 2D? How does this conversion work?

190 'simplest'

204 'it has its limitations'

223 '…that can enable the researcher to obtain an informed…'

235 'both figures'

In the example case studies, with such big clear outcrops, can you analyze a small area within the larger area and verify that you are accounting for the censoring correctly?

Recent reference of possible interest: Forstner, S.R., Corrêa, R., Wang, Q., Laubach, S.E., 2024. Fracture length data for geothermal applications. In Gill, C.E., Goffey, G., Underhill, J.R., eds., Powering the Energy Transition through Subsurface Collaboration, Geological

Society of London, Energy Geoscience Conference Series, v. 1,
https://doi.org/10.1144/egc1-2024-17

350 Given the limitations of any spacing statistic, I think it would be worthwhile mentioning here that good field practice with scanlines should be to keep track of the sequence of fracture occurrences, in other words, the spatial arrangement, as you've pointed out in other work (and also Marrett et al. 2018, J Struct Geol). Your analysis here seems like it would be equally apt for spatial arrangement data collection and analysis.

376 I agree with this way of proceeding re: defining length. Does your method work as well with lengths defined via branches; is there a reason to choose one or the other? Maybe this gets out of scope, but the way you mention it here might make a reader wonder.

388 This is a big claim that length is always underestimated. What if you have a process that produces only short fractures (or even fractures that are shorter than your outcrop size). Hooker et al. 2013, J. Struct. Geol. describes one set (of several) that only contains very short lengths. Maybe some caveats are in order here.

395 'it'? Maybe 'they are'?

405 Although testing this hypothesis is something that people studying fracture lengths in the context of geomorphology ought to consider. Particularly large or open fractures can affect the size, shape, and occurrence of outcrop. See Eppes et al. 2024, Earth Surface Dynamics, doi.org/10.5194/esurf-12-35-2024.

409 'in key of time' is an odd phrase. Check.

411 'useless' seems harsh. I'm not convinced this extra remark is needed. Anyway, there may be other parameters (like segmentation) that have similar effects to outcrop size that would benefit from the approach you propose, even if outcrops were arbitrarily large.

445 This assumes that measurements are only caried out at one scale of resolution. But this need not be the case. See Ortega et al. 2006, AAPG Bulletin (for aperture sizes) and Forstner et al. for lengths.

451 And for some fracture systems, the smaller fractures are more prone to be mineral filled and potentially less obvious features on images. This size/visibility effect can also manifest in the picking of long fractures if the long traces are segmented.

---

## Author Comment (AC1)

**81** I suggest making this comment more nuanced and adding a reference [...]

Thank you, we changed the line as suggested.

**106** Here the flow in the fracture network is assumed to be a function of connectivity, but in the preceding list of fracture types many of the elements many not be conducive of fluid flow, for example some faults and opening-mode fractures that are sealed. Likewise, if you have a situation where sets are of different ages, early sets may be sealed (or partly sealed) and later ones more open. An example is an outcrop of veins abutted or crossed by later joints. These abutting and crossing relations will have a different impact on flow than a bunch of intersecting joints. Maybe in say: "If all the fractures are open, a network with prevalence of I nodes..." This may not be central to the point that you are making in this paper, but it's such a common and misleading logical jump in fracture network studies that the clarification is useful. See the discussion in Forstner and Laubach, 2022, J. Struct. Geol. Also, if the rock itself is porous, even a network that has only I nodes can markedly augment fluid flow (Philip et al., 2005, SPE Res. Eval. Eng.), here length distribution is the key parameter (not connectivity) which just makes your focus on length even more important.

We agree on the importance of clarifying, we expand the line as following:

**106** In a non-porous rock with all open fractures, a network with a prevalence of I nodes will be less connected, and fluid flow will be more restrained. However, for many of the fracture types that were previously discussed, this is indeed not the case (e.g. sealed faults and opening-mode fractures) (Forstner and Laubach, 2022). Furthermore, if the rock is porous then length distribution becomes the key parameter for controlling fluid flow (Philip et al., 2005).

 It seems like these values might also be meaningless for 'stochastic modeling'? Do you clarify this in the Discussion?

The discussed values indeed are useful for stochastic modeling both from a statistical and numerical point of view (i.e. DFNs). In the text this was not explained clearly. We provide the following edit to clarify.

119 Circular scan lines methods on the other hand do offer an unbiased estimate of the mean length, however, being non-parametric, they do not yield neither the distribution type (e.g. normal, exponential, etc.) nor distribution shape parameters (e.g. standard deviation, etc.). This in turn, makes the estimate completely useless to quantitatively compare different results, and carry out any downstream statistical and/or numerical modelling, such as DFN stochastic fracture modelling.

129-132 On first read, I found the transitions here confusing. For clarity I think you ought to warn the reader here that you are going to demonstrate the time-length dimensional shift in 3.3. Something like 'Survival analysis is usually used in the time domain. In section 3.3. we show how a time-length dimensional shift is valid. Here we briefly introduce the terms as they are used in the time domain.' These are key lines defining terms. I think they could use some clarification. What do you mean by 'the even of interest is commonly defined as *death*'? Is a clarifying word missing? The 'event of "x" is'? Or do you need a some more information at the start of the paragraph: "Survival analysis is used to analyze data in which the time until the event is of interest (for example, the time until death in some medical or biological contexts)." This would perhaps be a good point to introduce the idea that you are substituting distance for time?

Thank you for pointing out that you found the transition confusing. We changed the text as follows hoping to make it clearer.

122 To solve these problems, we propose to use survival analysis, a specialized field of statistics, specifically developed to deal with censored data. Survival analysis focuses on the analysis of time of occurrence until an event of interest (Kalbfleisch and Prentice, 2002). The advantage of survival analysis over the methods discussed above is that it considers censored data as the carrier of the crucial information that the event did not occur up to the censoring time, thus allowing for an unbiased estimation of all statistical parameters and models. However, although in literature the terms survival times, time-to-event, or more generally lifetimes (Lawless, 2003) seem to imply that time is the only valid variable, any non-negative continuous variable, such as length, is valid (Kalbfleisch and Prentice, 2002;

Lawless, 2003). In the following sections of this chapter, we will start describing the canonical theory behind survival analysis in function of time, and then we will show how the same theory can be applied in space, to sets of length or distance measurements.

129 Since this technique is rooted in medical and biological applications, the nomenclature from this type of literature is carried along. The event of interest (for which we measure the time-to-event) is often defined as death, while a loss indicates that the observation has been lost because it was hindered by a secondary event, called a censoring event. Censoring can be ...

133 Which 'length' do you mean here?

Changed with:

133 the event happens after the end of the study period and thus we observe the partial lifetime of the event.

173 1D or 2D? How does this conversion work?

We did not understand if the comment is referred to the type of intersection between the fractures 173 4. the censored event as the intersection between the fracture trace and the boundary (marked by a B node), or if it is referring to the figure below indicating that it is not clear what the figure entails. For the former it is a 2D intersection. For the latter, then we can expand the figure caption and text as follows:

Figure 6. Censoring effect on an example of a simple fracture network and corresponding survival diagram. The survival diagram is a 1D representation of the fracture length. On the Y axis the fracture number is indicated and on the X axis the length is measured. Solid lines indicate the actual measured length while dashed lines indicate the possible continuation of the fracture. Yellow pentagons represent the censoring of the boundary.

174 Figure 6 represents an abstraction of the fracture network by just representing fractures by their length. Each fracture in the network is numbered (Y axis) and the

corresponding fracture length is represented by a bar. Bars with a yellow pentagon indicate that the fracture n is censored and thus the measured length is shorter than the true length. By applying …

190 'simplest'

204 'it has its limitations'

223 '…that can enable the researcher to obtain an informed…'

235 'both figures'

395 'it'? Maybe 'they are'?

409 'in key of time' is an odd phrase. Check.

411 'useless' seems harsh. I'm not convinced this extra remark is needed. Anyway, there may be other parameters (like segmentation) that have similar effects to outcrop size that would benefit from the approach you propose, even if outcrops were arbitrarily large.

Changed in the main text. Thank you for the corrections.

350 Given the limitations of any spacing statistic, I think it would be worthwhile mentioning here that good field practice with scanlines should be to keep track of the sequence of fracture occurrences, in other words, the spatial arrangement, as you've pointed out in other work (and also Marrett et al. 2018, J Struct Geol). Your analysis here seems like it would be equally apt for spatial arrangement data collection and analysis.

We added a brief mention of this in the text as such

371 Finally we would like to point out that the censoring analysis is a secondary part in the analysis for spacing. It is worth noting that analysing the spatial arrangement of the fractures in the network (such as Marrett et al. 2018 and Bistacchi et.al 2020) is of fundamental importance. The presented datasets are equally apt to this type of analysis; however, we decided not to include this analysis and focus mainly on censoring to avoid increasing the length of an already dense text.

376 I agree with this way of proceeding re: defining length. Does your method work as well with lengths defined via branches; is there a reason to choose one or the other? Maybe this gets out of scope, but the way you mention it here might make a reader wonder.

Yes, we chose to measure the lengths of the entire segments instead of branches because they entail two different things. Branches offer a useful topological abstraction of the network (making it possible to classify node intersections), but they do not have a real geological or physical meaning. As we defined in section 2, 2D fractures traces are the intersection of discontinuity surfaces with a secondary surface. Branches on the other hand are defined as a segment of a fracture trace between any two nodes (either I-I, I-Y etc..). Considering the geological origin of a trace, by using branches we would be essentially segmenting fracture planes in smaller sub-planes. This, however, is only an artifact given by the topological definition of a branch and thus the obtained branch length distribution does not carry any real physical meaning.

This discussion, as interesting as it is, may be a bit out of scope and we tried our best to summarize it in the discussion as follows:

376 Branches offer a useful topological abstraction of the network (making it possible to classify node intersections), but they do not carry a real geological or physical meaning and as such a distribution obtained by fitting branch-length will have a different meaning compared to a length distribution.

**388** This is a big claim that length is always underestimated. What if you have a process that produces only short fractures (or even fractures that are shorter than your outcrop size). Hooker et al. 2013, J. Struct. Geol. describes one set (of several) that only contains very short lengths. Maybe some caveats are in order here.

Yes, however we firmly support it and we expanded the discussion to motivate it further:

**423** Measured lengths of censored fractures will always be shorter than their true lengths and, by using the first simple approach, the dataset is essentially "polluted" by shorter fractures thus always decreasing the measured mean. The second simple method will also lead to an underestimation of the mean because of the size bias. However, this second method can be less impacted by censoring. For example, if a fracture population has a very small standard deviation (i.e. almost all fractures have the same length) and/or fractures are occurring in an outcrop that is much bigger than the characteristic fracture length, then removing censored values would not have a great impact on the estimation. But, even if small, the underestimation will always be present. Overestimation of the mean length would be possible in these scenarios when we do not consider censoring as independent from the length distribution (for example if only fractures shorter than a certain value are censored). However, this would violate both the core underlying hypothesis of random censoring, and standard geological experience, and thus we do not deem it possible under these imposed limits.

**405** Although testing this hypothesis is something that people studying fracture lengths in the context of geomorphology ought to consider. Particularly large or open fractures can affect the size, shape, and occurrence of outcrop. See Eppes et al. 2024, Earth Surface Dynamics, doi.org/10.5194/esurf-12-35-2024.

We added this remark in

**406** ... independent processes. Nonetheless in some applications (Eppes et al. 2024) this assumption may not hold, and a more in-depth study may be required to prove the independence hypothesis before proceeding.

445 This assumes that measurements are only caried out at one scale of resolution. But this need not be the case. See Ortega et al. 2006, AAPG Bulletin (for aperture sizes) and Forstner et al. for lengths.

Changed to

445 Because of this limitation, for **a constant resolution scale**, the modelled length distribution

451 And for some fracture systems, the smaller fractures are more prone to be mineral filled and potentially less obvious features on images. This size/visibility effect can also manifest in the picking of long fractures if the long traces are segmented.

Added as another factor contribution to censoring, thank you for the suggestion

In the example case studies, with such big clear outcrops, can you analyze a small area within the larger area and verify that you are accounting for the censoring correctly?

This is a tricky question that we thought about while writing the paper. It is not easy to see if censoring is correctly accounted for by just subsampling the outcrop (as large as it is). The problem is that essentially, we do not have a controlled environment. First, we are estimating only a limited suite of statistical models and thus we cannot say if the best estimated model is the true underlying model (and in fact we will never be able to tell). So even if for the first case study the lognormal may seem perfectly fitting, we cannot be certain that it is the true underlying statistical model. Moreover, the spatial distribution of the fractures in the outcrop space is also not uniform thus with the same sub area dimension you will have different model estimations depending on the position of the sub area. Thus, we found it difficult to obtain a satisfactory estimate of how well censoring is accounted and how well survival analysis works depending on the censoring percentage. We are relying on the fact that survival analysis has been used and is still being used in countless applications and show that it is working also for lengths. However, we believe that synthetic experiments can and should be carried out to explore further the effects of censoring, violations of the underlying hypothesis on the final estimation and the overall precision and reliability of survival analysis (we talk about this in 407-435). We decided to not include or explore synthetic results because it would have drastically increased the length of the MS and blurred its focus.

---

## Author Comment (AC2)

ABSTRACT

Abstract generally: I feel a lot of this amazing research is not represented well enough and gets lost in a rather unstructured paragraph.

We followed the suggestions by also including the contents of the lines 56 and 58 of the introduction as suggested. The abstract is now as follows:

- Line 3: Too vague. What and who's necessities shifted and why now? Who demands parametric solutions?

- Line 4: Too vague. Please rephrase "parametric solutions to compare […] all the parameters". Avoid doubling word stems (parameter) and avoid being too general ("all the", "several types").

    Changed to (lines 3-4):

    However, due to the recent raise in popularity of Digital Outcrop Models (DOMs) and of stochastic Discrete Fracture Networks (DFNs), there is an increasing demand for distribution-based solutions that output a correct estimation of parameters for a given proposed model (e.g. mean and standard deviation).

- Line 5: Word order swap: Move "in geological literature" backwards. Work with the "absence of [something]" rather than interrupting this segment.

- Line 5: In line 3 present tense was used, now past tense. I recommend present tense here. Further: "These changing requirements" is fine to say once they are specified, else the link is lost (see comment line 4)

    Changed as such:
    This change in demand highlights in geological literature the absence of properly structured theoretical works on this topic.

- Line 6: When using "in particular" I recommend using "in general" beforehand to tie the parts together. Again, missing links in storyline.

- Line 7: At this point the reader does not necessarily know about right and left censoring, as only later explained in line 133. Either explain here or mention that it is a specific type of censoring that will be explained later in detail. Further: Mark word in italics?

    Changed lines 6 to 7 as such:

    Our methodology, presented for the first time in this contribution, represents a powerful alternative to non-parametrical methods, aimed to specifically treat censoring bias and obtain an unbiased trace length statistical model.

- Line: 11: I don't understand "modified" in this context or location. Further, please avoid long sentences; split this sentence into two sentences?

    Changed as suggested:

    As a second objective we propose a novel approach for selecting the most representative parametric model. We combine a direct visual approach and the

calculation of four statistics to quantify the distance between the proposed and the true underlying model.

- Line 12: How often or on how much data? Maybe place the term "correctly" more prominently; it gets a little lost while this is the main selling point!

Modified as such:

Finally, we apply survival analysis to correctly estimate statistical parameters of censored length dataset in three different case studies. We furthermore discuss the effects of censoring percentage on crude parametrical estimations that do not use this paradigm.

INTRODUCTION

- Line 16: composed of

Changed as suggested

- Line 19: Replace "Nowadays the increase in" with "Amplified"? Wording sounds dated. Does non-Italian research in DOMs exist? It feels biased. Try being more diverse in reference selection.

- Line 21: Grammar: Plural. DOMS allow the extraction of datasets? Try being more exact in wording.

Changed 19-21 as such:

19 The recent increase in computing power and the emergence of new approaches based on Digital Outcrop Models (DOMs) allow for the extraction of large datasets and facilitate the measurement of properties instead of just their estimation (Bistacchi et al., 2015; Tavani et al., 2016; Healy et al., 2017; Thiele et al., 2017; Marrett et al., 2018; Nyberg et al., 2018; Bistacchi et al., 2020; Martinelli et al., 2020; Bistacchi et al., 2022; Mittempergher and Bistacchi, 2022; Storti et al., 2022).

- Line 32: "only indirect geophysical methods may provide truly 3D datasets". Truly as adverb to provide? Or true as adjective to 3D datasets? Change word order or grammar. Further, is there a better way of saying it?

Changed "truly" with "complete"

- Line 34: "a rich literature" sounds odd. Try "vast research conducted" or else.

- Line 34: Please don't start sentences with "Because of". Try "Due to" or "Given" or else. Please follow academic writing guides.

- Line 35: "the 2D lines of intersection of 3D [...] surfaces with the outcrop surface, or with topography" is too complicated. Please make sure to keep sentences short and clear.

We agree that the phrase is too long. We have applied the suggestions as such:

34 Due to this, a vast body of research focuses on the characterization of properties of discontinuity traces or lineaments, i.e. the 2D intersections of 3D discontinuity surfaces with the outcrop surface, or with topography (Dershowitz and Herda, 1992; Bonnet et

al., 2001; Manzocchi, 2002; Bistacchi et al., 2011; Bistacchi et al., 2011; Sanderson and Nixon, 2015; Martinelli et al., 2020; Storti et al., 2022).

- Line 41: "the Digital Outcrop approach". Is this a standardised method? It has not been mentioned in the text before. It is also not explained in more detail in the following sentence as the reader might expect. Please link the sentences more carefully and guide the reader better. Please consult academic writing guides.

We maybe understand the confusion. The wording may confuse the reader since it is not clear that we mean that DOMs are the tools used and not the methods for defining unbiased lengths. We rephrased as such:

39 In this contribution we focus on the problem of defining accurate and unbiased length (or height) distributions based on data collected on DOMs.

- Line 43: don't capitalize "authors"

Changed

- Line 44: "consists of"

Changed

- Line 47: Why reference here and not at the end of the sentence?

Modified

- Line 50: Here: DOM approach. Whereas in Line 41: Digital Outcrop approach. Please avoid using multiple versions of one term.

Changed to DOM

- Lines 50-52: Please don't put words in bold. They have no meaning individually here and the reader is able to read text just so. They stand out too much in light of the text here.

Removed bold

- Line 53: Please avoid disclosing information in brackets. Instead, convert them to separate sentences.

We have changed as follows (accommodating also the suggestion from Healy):

48 Thanks to its simple implementation in the field, this technique is widely used popular however it has an important limitation: lineament lengths are never directly measured. Due to this, analysis carried out with the circular scanline method yielded estimates of mean values without a complete characterization of the lineament length distribution and without any real statistical significance.

- Line 66: time spans. Also, please do not put in bold.

Changed

- Line 67: length measurements

- Line 69 "that is the main topic of this contribution" is a separate, full sentence. Do not add this to the previous sentence. Please check academic writing guides. What is physical measuring? Please clarify.

- Line 72: Avoid "second objective" if there has been no "first objective". Please re-read academic writing guides. Always stay consistent.

- Line 73: These hypotheses come as a surprise, as they have neither been introduced nor are they explained here. Please adjust.

We have changed the text following the suggestions of the lines 67-73 as follows:

67 Measuring length is straightforward with dedicated code or with a simple GIS software however, applying survival analysis and fitting robust unbiased parametric statistical distributions, needs a more detailed treatment. As the main topic of this contribution, we propose to adapt survival/reliability analysis techniques to correctly account for censored lengths and estimate robust trace length distributions derived from DOMs. Furthermore, we define a quantitative methodology to select the most representative estimated statistical model (i.e. parametric distribution) from a list of proposed models.

The theory and techniques presented in this paper are available as an open-source Python package called FracAbility that accepts shapefiles as input and allows to carry out a complete and unbiased statistical analysis workflow for fracture length data (https://github.com/gecos-lab/FracAbility).

FRACTURE SURVEYS AND TERMINOLOGY

Overall: I disagree with the writing style. It seems written almost as if it was spoken language. Often too many thoughts are cramped into one sentence, sometimes not clearly separated. Using parentheses to introduce even more points must be avoided. Important points need to be moved to the beginning or end of the sentence or paragraph. Please make sure that academic writing advise is followed thoroughly.

We understand the disagreement and the comments on the style. We have followed the suggestions and tried to improve the text. We have also changed the order of the contents of this section to better accommodate the suggestions and improve readability. Here we only provide the changes referenced to the comments.

- Line 79: Language is imprecise. Joints are never "empty". There will always be fluids/gases if not solids. Better: lacking mineral filling, or else

Previously modified with the comments provided by Laubach but we have expanded the contents of the reviewer's answer to be clearer:

78 Fractures are classified as shear or tensional fractures. Tensional fractures, when possible, can be further divided into joints when empty or veins when filled by minerals (Twiss and Moores, 2007; Davis et al., 2012). This distinction however is often difficult to make without an in-depth field and sample validation since many fractures have hybrid fill attributes (Laubach et al., 2019)

- Line 85-86: Avoid making multiple points in one sentence. It exhausts the reader too quickly.

- Line 88-90: Split in two sentences please.

- Line 94: Please rearrange the sentence order. Think about what point is trying to be made and move it either to the front or end of the sentence. Please check academic writing guides for this.

- Line 94: "impossible" is too emotional. Use "not feasible" or else.

- Line 95: Never use "…" in an academic manuscript.

- Line 91: Does the sampling area reach from a thin section to a satellite image? Language is not precise enough. Please improve.

- Line 92: Please do not put new points in brackets. Either make them a new sentence, if important, or leave them completely out.

- Line 93: Why "boundary nodes" in bold if not "nodes and branches" (line 98) in bold as well? Please stay consistent. Generally, avoid bold style.

- Line 100: If it is not a direct quote there is no need for a page reference

We have changed the text from line 84 to line 100:

84 Although three dimensional by nature, most of the times discontinuities appear as 2D lineaments or traces over a surface. These lineaments are the intersections of such discontinuities with a secondary surface, such as the topography, an outcrop, a borehole or a sample. In this work, following a common usage in outcrop studies, the term *fracture* is used as a generic term to indicate any type of discontinuity trace. Fractures with the same formation age, kinematics and orientation, can be grouped into families or sets. Multiple fracture sets present within an area, form a fracture network or system (Davis et al., 2012). Additionally, fracture networks present two other main components: *Boundaries* and *Nodes* (Figure 1A). The *boundaries* are the limits of the sampling area, that can be at the scale of a thin section or a satellite image, within which the sampled fracture traces are assumed to be complete. Ideally boundaries are strictly convex however this is often not the case. Boundaries often show tight bends, coves and holes and the final shapes are mostly controlled by localized alteration, anthropogenic activity, vegetation etc. (Figure1B).

*Nodes* are points in the network that define how fractures interact or don't interact with other elements of the network (other fractures, boundary, holes). Nodes can be classified by the number and type of segments (branches) that insist on the given point (Sanderson and Nixon, 2015):

....

- Line 106/107: Why future tense and not present tense? In line 110 present tense is used. Please stay consistent.

  Changed as follows (we also included the change from Laubach)

  In a non-porous rock with all open fractures, a network with a prevalence of I nodes is less connected and thus fluid flow is usually more restrained. Conversely, both Y/T and X nodes increase the connectivity of the network and thus increase both fluid flow and the permeability.

- Figure 1: Maybe mark the "hole" with red hatching? This makes it easier to find the "hole" quickly and follow the explanation.

  Changed as suggested

- Figure 2: "Topology" seems to be spelled with an odd character. Number 8 in the right-hand picture seems dislocated.

  Fixed

STATISTICAL MODELLING OF CENSORED LENGTH DATA

General: The manuscript needs a lot more structure at this point to avoid the impression of a random collection of thoughts. Introductory sentences of paragraphs need to start with catchy topics. Final sentences need to sum up the information or conclusion. One thought per sentence only. Keep sentences short. Start and end of paragraph need to communicate with one another.

We have followed the proposed suggestions to improve the text

- Line 114/115: The start of the sentence and paragraph is poor. Try "There is increased necessity for estimating parameters of statistical distribution in length datasets". Before "however" there should be a full stop. First person plural should be avoided.

- Avoid putting all but one word of a sentence in bold. It looks accidental. Maybe introduce the question with a colon or just don't make it bold.

- Line 119: "lines" = line

- Line 120: Do not write "on the other hand" if there is no "on the one hand".

- Line 122: "almost completely meaningless and useless" is too emotional. Use "impractical" or else.

- Line 124-126: This sentence makes no sense to me. Please rewrite.

114 Having a length dataset, there is usually the necessity of estimating the parameters of one or multiple statistical distributions. When doing so, censoring is inevitable as the boundary in which measures are carried out will always be limited. Then how can we carry out an unbiased estimation of such parameters? On one hand, common and simplistic approaches such as ignoring censoring (i.e. considering censored lengths as if they were complete) or excluding censored measurements (i.e. cherry-picking only non-censored data) should be avoided as they will always lead to an underestimation of the model parameters (see discussion for a more in-depth analysis). On the other hand, circular scanlines methods offer an unbiased estimate of the mean length, however, being non-parametric, they do not yield neither the distribution type (e.g. normal, exponential, etc.) nor distribution shape parameters (e.g. standard deviation, variance, etc.). This in turn makes the estimate's use-case quite limited and not apt to possible statistical modelling applications. To solve these problems, we propose to use survival analysis, a specialized field of statistics, developed to deal with censored data. Survival analysis focuses on the analysis of time of occurrence until an event of interest (Kalbfleisch and Prentice, 2002). Although in literature the terms survival times, time-to-failure, or more generally lifetimes (Lawless, 2003) seem to imply that time is the only valid variable, any non-negative continuous variable, such as length, can be valid (Kalbfleisch and Prentice, 2002; Lawless, 2003). The advantage of survival analysis over the methods discussed above is that it considers censored data as a valid datapoint, carrier of the information that the event did not occur up to the censoring time. This is a necessary shift in perspective that allows for an unbiased estimation of statistical models that will be described in the following sections of this chapter. We will start by describing the theory behind survival analysis in function of time, and then we will show how the same theory can be applied in space, to sets of length or distance measurements.

- Lines 133-140: Always start with a capitalized letter after a colon.

  Changed

- Line 134: Why first-person plural here? Please avoid. Use passive voice. Maybe use: "the event happens after the end of the study period and thus the length of the event is partially observed".

  Changed to: "The event happens after the end of the study period and thus the partial lifetime of the event is observed;"

- Lines 135-140: Follow advise from line 133.

  Changed (capitalization)

- Lines 142-145: Avoid writing in bold.

  Removed bold

- Line 145: time to failure or time-to-failure? Keep spelling constant. Also, this term was only mentioned in an example and inside brackets – explain this more in detail instead.

  Changed to "lifetime"

- Line 150: I understand "complement" as a supplement or accessory. Is "inverse" meant?

Changed to "opposite"

- Line 166: "non-negative continuous variable" is not defined and is not the opposite to "valid variable" yet seems to be the "central point of this work". The term needs more introduction if it holds such importance.

This was moved in the introductory paragraph of the chapter (comment for lines 114-126):

Although in literature the terms survival times, time-to-failure, or more generally lifetimes (Lawless, 2003) seem to imply that time is the only valid variable, any non-negative continuous variable, such as length, can be valid (Kalbfleisch and Prentice, 2002; Lawless, 2003).

Thus line 166 becomes:

166 Since length is, as time, a non-negative continuous variable, it is theoretically possible to apply survival analysis techniques to length datasets by considering (Fig. 6):

- Line 170: Why highlight a verb if else only nouns are put in bold? Stay consistent by changing wording or marking.

Removed bold for the verb also changed to lifetime

- Line 174: What are "the definitions of the different types of censoring"? They are not mentioned. The bullet point list mentions considerations. Why are definitions referred that are not mentioned in this subchapter?

We provided a link to the section where the types of censoring are described.

- Line 176: Why are sources of fracture genesis mentioned here in this subchapter? Is this not a topic for the introduction?

The sources in parentheses are related to secondary events not to fracture genesis, we changed the text as follows:

174 By applying the definitions of the different types of censoring (described in 3.1) to our specific application, it is reasonable to assume that only random censoring occurs in trace length analysis. Moreover, censoring is non-informative since the boundary is the product of secondary events (i.e. alteration, erosion, debris covering parts of the outcrop, vegetation, human activity, etc.) that occur after the fracture genesis and thus do not inform the occurrence of the event (see the discussion for a more in-depth analysis)

- Line 189: If a colon is used to present the main objective, avoid listing side points as bullet points. The message gets lost, and the reader is confused.

  Changed as follows:

  184 Given a statistical parametric model with density g(x, θ) (i.e. an assumed theoretical distribution) and a sample x of size n, the main objective of MLE is to estimate the parameters $\hat{\theta}$ such that the observations x are the most likely under g(x, $\hat{\theta}$) (Burnham and Anderson, 2002, 2004; Karim and Islam, 2019)

- Line 190: simples to simple

  Changed to "simplest"

- Line 196: Avoid putting words in bold

  Removed bold

- Line 209: uncertainty to uncertainties

  Changed to "uncertainties"

- Line 212: What are natural questions? Please clarify.

  Removed natural

- Line 213: Changing several short simple questions to a long complicated question is not "reducing". Starting a question with a side sentence is not recommended. Structure needs to be clearer.

  Changed as follows:

  With a series of fitted models it is important to understand which model better represents the data. In literature this is usually archived using a specific type of null hypothesis tests defined as goodness-of-fit (GoF) test (Storti et al., 2022).

- Line 216 ff: Do not use first person plural. Instead: "These types of tests…"

  Changed as indicated

- Line 224: "Sensible" sounds highly subjective if not explained how this is defined.

  Line 256: "models deemed reasonable by the researcher" again is very subjective. Can this be made more objective?

  We added the following to address these comments:

  224 By sensible candidates it is intended those models that make physical sense for the case study. For example, for fracture lengths, a power law can be considered a sensible candidate since it describes an observed pattern of fracture self-similarity. A lognormal is also an equally sensible candidate because of the effect of truncation. Conversely a normal distribution is not very sensible since values can be negative. Thus, for a normal model, either the average length is very high and standard deviation very low or a truncated normal should be tested. In the case studies section, we will cover different models purposefully adding non-sensible models (such as a standard normal) to showcase the model selection workflow.

- Line 226: Do not start a sentence with rather meaningless introduction words. The reader's interest is immediately lost. Instead start with the subject "Probability Integral Transform is […]"

- Lines 228-231: Do not name the conditions (list) if the transformation statement is promised after the colon. Instead write line 231 first after the colon. The list gives too much importance to the conditions and limits the focus on the actual definition.

Changed 226-231 as following

228 The Probability Integral Transform (PIT) is a well-known visual transformation of continuous distributions which states that the frequency of Z is distributed following a standard uniform (Fisher, 1930) given:

....

- Line 233: Split sentences after "(Fig. 7A)". One thought per sentence only.

Added a full stop after fig7A

- Line 248: likelihood of...

Added "... the model"

- Line 295: Again, important messages need to be at the start or end of the paragraph. This is not the case here. Make sure "sensible guided choice" is put in the end.

Changed as suggested:

293 By comparing multiple rankings, even if different distances are ranked in differently, it is still possible to make a sensible guided choice, for example by using the PIT representation together with the mean ranking position or follow a specific type of distance.

- Figure 3 and line 136: "observations intervals" = observation intervals". A figure should always speak for itself. This figure does not make enough sense by itself and needs simplification. Reduce the number of colours, avoid unnecessary and unlisted abbreviation (e.g. start and end), make sure colours can be distinguished (e.g. black vs. grey; tightly-dashed vs. line), explain question marks, match thickness of lines in image and legend, standardise spacing behind "Complete" and "Interval", standardise font size for y-axis title, y-axis is not an axis, etc.

- Figure 4: standardise axis titles font sizes, match image and legend line thickness, explain "C" in figure, clarify definition for partial length, use "s" and "r" in the image if explained within the legend, A and B should be on the top left side of each image, full-stop missing in figure description.

- Figure 5: Standardise font size for A and B, make box clear unless figure B is red, increase all axis title and value font sizes, remove title for figure or increase font size, define axis title x.

- Figure 6: Match axis title font sizes to legend font size, Left: Match legend to image. Right: Match line thicknesses, clarify "n." on y-axis: title number would be abbreviated "no.", why clarify unit of length but not item (fracture length)?

- Figure 7: The numbering (A, B, C) would normally occur on the left-hand side of the sub-figures. The text appears rather small and might be increased – however the message is clear.

Improved images following the proposed suggestions

CASE STUDIES

- Line 297: "all the discussed theory" to "the discussed theory"

Changed to "the discussed theory"

- Line 298: In the sentence before it is "case studies" so one needs to introduce singular first: change "one" to "case study"

Changed to "the first case study"

- Line 307: Try to mirror sentences that belong together. If starting with "Pictured on the left…" continue similarly to guide the reader. Don't say "On the right it is represented" but try "Pictured on the right" or similar.

Changed to: "Pictured on the right, a subarea of the quarry with the boundary, fracture traces and boundary nodes (yellow pentagons)"

- Line 327: "Weibull seem" to "Weibull model seems"; "as" to "than"

- Line 328: Again, the last sentence has little value in the paragraph ("occupy the last two positions"). Maybe better: "rank lowest in comparison of the distances/models" or "are least representative".

Changed both as suggested:

323 For the calculated distances scores, the lognormal is followed by the gamma and Weibull models, however the rank scores of these last two models are not uniform. While the Koziol and Green distance favours the gamma model, the Anderson-Darling distance favours the Weibull model. Finally, the power law and normal models rank lowest in comparison to the other models indicating that they are less representative.

- Line 336, 337: Why past tense here when nowhere else?

Changed as following:

336 This leads to an inevitable deformation of the orthophoto (and thus of the measured lengths) along the extremities of the analysed area.

…

The analysed fractures are subdivided in two main sets, striking NE-SW (Set 1) and NW-SE (Set 2), conforming to the general trend of the area for brittle deformation (Bistacchi et al., 2000; Bistacchi and Massironi, 2000; Bistacchi et al., 2001).

- Line 341. Do not start a paragraph and/or sentence with the least important information ("In Fig. 12").

- Line 341/342: Can this sentence be smoother?

Changed as follows:

341 The different estimated models are represented for both sets in Fig. 12. In both cases the lognormal model, overall, better fits the data.

- Line 346: replace "afterwards" as it is too figuratively. Try "at greater lengths" or else.

Changed with: "for longer length values"

- Line 349: "For the other models, looking at the mean rank value helps in understanding the final ranking showing that the gamma distribution is ranked lower than the exponential and the Weibull (at the second and third place respectively)" reads very clunkily. Try and rephrase.

Changed as such:

For the other models, the mean rank value can be used to assign a final ranking. The mean ranks show that the gamma distribution is less representative than both the exponential and the Weibull (respectively the second and third most representative models).

- Line 350: Case study 1 and 2 are locations whereas case study 3 is a topic. Maybe make that clearer by expanding the title, e.g. "Spacing analysis"

Changed subsection title as suggested

- Line 351: Do not start a paragraph or sentence with minor information like here. Start with "Survival analysis can be used…"

We have changed as suggested and also moved a suggestion from the Laubach review to improve the structure of the section (previously it was at the end of the section).

351 Survival analysis can be used also to analyse the spacing length distribution for each fracture set. Thus, the same workflow was applied to spacing measurements of both S1 and S2 of the Colle Salza dataset.

It is worth noting that censoring analysis is a secondary part in the analysis for spacing. Analysing the spatial arrangement of the fractures in the network (such as Marrett et al. 2018 and Bistacchi et.al 2020) is of fundamental importance however we decided not to include this analysis and focus mainly on censoring to avoid increasing the length of an already dense text.

- Line 366: Start new sentence with/at "Thus"

Changed as indicated

- Line 367: Add "model" to end of first sentence. "bad" to "poor".

- Line 368: Don't use "place". Try "rank" or else.

Changed as follows:

367 The poor fit of the lognormal model is also confirmed by the other distances that rank the lognormal lower than the Weibull and gamma models (ranking first and second respectively).

- Line 370: Do not use "on the other hand" if you haven't used "on the one hand". Mind structure and consistency please.

- Line 371: What does "converging all at the first place" mean?

- Line 370/371: Repeated use of word "quite"

Changed as follows:

370 Conversely, the lognormal model is clearly the most representative for set 2 (Fig 15B). The PIT plot showing a quite linear behaviour and all the calculated distances are ranking first.

- Figure 8: Text is very small. "Yellow pentagons" can't be identified at this scale. Second scale in left figure should be inside the zoom window.

- Figure 9: Remove title or make bigger, reference length double units name standardized/ true.

- Figure 10: Title is too large, table is too small. Numbers and axis titles are small and hard to read. I suggest putting numbering of subplots (A-D) on the left-hand side of each image. I would get rid of all articles to make the figure description consistent. Overall title: "plots"?

- Figure 11: Numbers and words need larger font. Insert scale of subplot left inside subplot box.

- Table 2: "indicating a worst fit in respect of the exponential and Weibull distributions positioned in second and third place respectively" reads very clunkily. "a worst fit" does not exist and "places" are only handed out at races. Try and rephrase.

Changed table caption as follows:

Models' distance and rank tables for both Colle Salza sets. The closer to 0 the better. For this length dataset, the lognormal is the most representative of the data for all the different distances. Using the mean rank, the lognormal is followed by the exponential, Weibull and gamma distributions. As with the first case study the Power law and normal distribution are the less representative.

- Figure 12: Increase font size of title and axis title. A and B should be on left-hand side of figure and potentially smaller (compare with other figures). The text could read better. Try "PIT visualization for the proposed length models is shown for Set 1 (A) and Set 2 (B) of the Colle Salza dataset. The red line represents the reference U(0,1); the closer a model's line is to this reference, the more representative the model. Among the models, the lognormal distribution demonstrates the closest fit to the reference line in both sets, although its fit is inferior to that observed in the first case study. Across both sets, all estimated models exhibit less linearity, with notable underestimation between 0.34 m and 1.5 m in Set 1 (A) and between 0.44 m and 2.57 m in Set 2 (B)." or similar.

- Figure 13: Reduce title font, increase table font, change numbering A-D to left-side please. Increase axis description. Here, all articles are kept consistent in the description. This is better than in Fig. 10.

- Figure 14: Reduce A-C numbering (compare with other figures). Increase legend. Explain all lines in legend. Purple colour hard to see. Stay consistent in your phrasing: Delete "Shows".

- Figure 15: Reduce numbering letter A and B and title, compare to other figures. Increase y-axis and clarify. Please correct figure description by the advice given above.

- Figure 16: Please adjust sizes referring to similar figures and comments above.

  Changed all figures as suggested

DISCUSSION

- Line 374: Please don't put information in brackets

  Changed by removing the brackets:

  With this work we delineate a robust statistical framework to quantitively analyse and statistically model fracture trace length  and spacing

- Line 384/385: Repeated use of word "useful"

  Changed 384 as:

  Furthermore, having an unbiased statical model is also extremely  important for engineering  applications.

- Line 407: Please only mention a "second point" if a "first point" was explicitly named (line 378 names a "crucial" point; maybe write "The first and crucial point"?

  Changed by adding: The first and crucial point

- Figure 17: Increase/ add axis titles and number. Increase titles (compared to numbering (A-D)). Increase legend size.

  Changed figure as suggested

---

## Author Comment (AC3)

Line 21: 'allows the extraction of large datasets and facilitates the measurement of properties' – these are still just samples from the population though; they are not the 'right' answer.

Changed with "measurement" with "sampling"

Line 50: note that FracPaQ does not in fact use the mean length statistic from these measures, just Intensity and Density; length statistics are calculated directly from the sample lengths. In addition, FracPaQ employs MLE methods to estimate optimum length distributions, with a Goodness of Fit approach. The work of Rizzo et al., (Rizzo, R.E., Healy, D. and De Siena, L., 2017. Benefits of maximum likelihood estimators for fracture attribute analysis: Implications for permeability and up-scaling. Journal of Structural Geology, 95, pp.17-31.) is not cited here, and it needs to be. As the current text gives an incorrect impression of FracPaQ functionality, I respectfully ask for clarification on these points.

Line 383: again, as above length stats estimation in FracPaQ does not use circular scanlines; we use the mean and standard deviation of the sample data and, optionally, MLE. Please correct this misleading statement.

Changed the following lines:

| Before | After |
|---|---|
| 43 Authors is based on circular scanlines (Mauldon,1998; Zhang and Einstein, 1998; Mauldon et al., 2001; Rohrbaugh et al., 2002; Healy et al., 2017). | 43 Authors is based on circular scanlines (Mauldon,1998; Zhang and Einstein, 1998; Mauldon et al., 2001; Rohrbaugh et al., 2002). |
| 48 Thanks to its simple implementation in the field, this technique became popular and, thanks to its computational efficiency and apparent simpleness, was also implemented in modern applications such as FracPaQ (Healy et al., 2017) to be used with the DOM approach. However, this method has an important limitation: lineament lengths are **never directly measured** and so the circular scanline method yields estimates of mean values **without** a complete characterization of the lineament length distribution and **without** any real statistical significance (e.g. variance can be estimated only under very limiting assumptions, Pahl (1981)). | 48 Thanks to its simple implementation in the field, this technique is widely used however it has an important limitation: lineament lengths are never directly measured. Due to this, analysis carried out with the circular scanline method yielded estimates of mean values without a complete characterization of the lineament length distribution and without any real statistical significance. |

| | |
|---|---|
| 54 Moreover, calculating the length and estimating any distribution other than the exponential, was difficult and computationally intensive (Baecher and Lanney, 1978; Baecher, 1980) and, due to limitations in early algorithms used to generate stochastic fracture networks, there was no real interest in estimating precise distribution parameters. | 54 Moreover, calculating the length and estimating any distribution other than the exponential, was difficult and computationally intensive (Baecher and Lanney, 1978; Baecher, 1980) and, due to limitations in early algorithms used to generate stochastic fracture networks, there was no real interest in estimating precise distribution parameters. These limitations are less present today due to the increase of computing power and thus new tools and techniques based on Maximum Likelihood Estimation such as FracPaQ (Healy et al., 2017; Rizzo et al.,2017) are readily available, enabling researchers to apply quantitative statistical inference on dense digitalized dataset. |
| 381 Because of this reason, non-parametrical methods such as those proposed by Mauldon et al. (2001) and implemented in software such as FracPaQ (Healy et al., 2017) are unfit since they are not linked to any model. | 381 Due to this, non-parametrical methods such as those proposed by Mauldon et al. (2001) are unfit since they are not linked to any model. |
| 385 With a parametric model this can be easily estimated by checking the length values associated to a probability chosen depending on the safety margin that is needed for the use case. Approaches such as ignoring censoring or removing censored data do provide a statistical distribution, however the censoring bias is still present and thus the results are skewed, always underestimating length | 385 With a parametric model this can be easily estimated by checking the length values associated to a probability chosen depending on the safety margin that is needed for the use case. Modern alternatives that use a simple implementation of MLE such as FracPaQ (Healy et al., 2017) are a good step forward however the censoring bias is still present and thus the results can be skewed, underestimating length |

Line 118: 'avoided at all costs' is a bit too dramatic; delete.

Changed to "avoided"

Line 123: 'completely meaningless' – again, too strong; with no other alternative, it can be a useful estimate, albeit limited.

119 On the other hand, circular scanlines methods offer an unbiased estimate of the mean length, however, being non-parametric, they do not yield neither the distribution type (e.g. normal, exponential, etc.) nor distribution shape parameters (e.g. standard deviation, variance, etc.). This in turn makes the estimate's use-case quite limited and not apt to possible statistical modelling applications such as stochastic DFNs.

Line 405: not sure this statement is true. Many outcrops are bounded by fractures; thus, the modern day process that has defined the boundary HAS been influenced by the geological structure and fabric of the rock mass.

We understand that the statement as is written is confusing and not necessarily true. We have rewritten this part to include a clearer explanation of independence (including a new part suggested by Laubach's review):

400 To correctly classify censoring as random, we must assume independence between the censoring and length distribution. By "independence" it is intended that the mechanisms behind the generation of a fracture length distribution is different from the mechanisms that censors such lenghts. The boundary, which represents censoring, is usually the product of secondary events that occur after fracture genesis (i.e. alteration, debris hiding part of the outcrop, vegetation, human activity, etc.). Thus, albeit it is often the case that such events are controlled by preexisting structures, the physical processes that caused censoring are not the same that generated the fracture set and thus the original length distribution. This leads to an important implicit caveat where the measured lengths must be related only to the mechanism that we are interested in modelling, for example lengths that are surely linked to tectonics and no other secondary events. Such discussion highlights that the assumption of independence is difficult to rigorously prove since the true distribution of the length of fractures it is not known (we only observe a set of complete and censored data). In some applications (Eppes et al. 2024) this assumption may not hold, and a more in-depth study may be required to prove the independence hypothesis before proceeding. Nonetheless, we believe that it can be safely assumed in geological applications when the appropriate field work and a posteriori analysis are carried out.

Line 470 – 'proper' – replace with 'better'.

corrected

Line 485 – regarding DFNs (and elsewhere in the ms); there are other approaches to modelling fractured rock volumes, for example effective methods and tensorial approximations. It would be better to mention and acknowledge these alternatives. DFNs are just one approach, among many.

We added the following lines in the introduction:

30 It is worth noting however that DFNs are not the only viable approach to model fractured rock volumes. Other methods such as tensorial approximations (Suzuki et al 1998, Brown and Bruhn 1998) based on the crack tensor measure (Oda 1989) are also present and quite used (Healy et al 2017 ).

Fig 5, 9, 12, 15, 17, – make the axis labels (numbers and text) bigger relative to the figure; hard to read.

Changed also following the suggestion from Weihmann's review.

---

## Author Response (AR1)

We thank all the reviewers that took the time to read and took part in the discussion. Here we will report the changes made to the main text following each reviewer's suggestions. Since there are some time gaps between the submission of the reviews we will report the changes in order of submission. In case of overlap (i.e. same suggestion) we will refer to the review that first suggested the change. To indicate the new applied changes, we use the line numbers of the new version of the MS and in brackets the line number in the diff file. We have also expanded parts of the text to provide more precise descriptions and explanations in section 3.5. We provide such additions as a final chapter in this document.

Blue text are the comments of the reviewers, in black comments from us and in red the additions/modifications that we made in the main text.

**Stephen Laubach**

Line 81: I suggest making this comment more nuanced and adding a reference: 'joints when empty or veins when filled (refs), although many fractures have hybrid fill attributes: they may be partly filled with inconspicuous mineral deposits that resemble joints, or the degree of fill may depend on fracture width, so that small fractures resemble veins (e.g. Laubach et al., 2019).' After all, many fractures of interest to subsurface applications are strictly speaking neither 'joints' nor 'veins'. In some populations small fractures are fill but wide fractures are open with a thin mineral lining. The old joints versus veins terminology is not helpful, and this is particularly germane for the discussion of length since 'open' fracture length may depend on these width-dependent mineral infills. It's better to call them 'opening-mode fractures or faults' and separately specify the fill state. Laubach, S.E., Lander, R.H., Criscenti, L.J., et al., 2019. The role of chemistry in fracture pattern development and opportunities to advance interpretations of geological materials. Reviews of Geophysics, 57 (3), 1065-1111. doi:10.1029/2019RG000671.

We changed the line as suggested:

84 (110) This distinction however is often difficult to make without an in-depth field and sample validation, since some fractures may have hybrid fill attributes and may be only partly filled with inconspicuous mineral deposits and thus resemble joints. Moreover, the degree of fill may depend on fracture width, so that small fractures resemble veins (Laubach et al., 2019).

Line 90: The ambiguity of lengths where, as is the common case, fractures are segmented and en echelon, ought to be mentioned. This is a big source of uncertainty in measured lengths (and heights) and there are now ways to deal with this rationally with other node types. See the Forstner paper.

We added this in the discussion:

396 (484) Furthermore, length can sometimes be affected by a subjective bias leading to the uncertainty of which structure should be measured. For example, with segmented or en echelon fractures do we measure the single segments or consider their cumulative sum? By approaching this problem with branches, a clear objective answer cannot be reached while by

considering a 2D lineament as a representation of the 3D geological structure, contextualized in the geological history of the outcrop, a more rational and informed decision can be made (i.e. Forstner and Laubach, 2022).

Line 106: Here the flow in the fracture network is assumed to be a function of connectivity, but in the preceding list of fracture types many of the elements many not be conducive of fluid flow, for example some faults and opening-mode fractures that are sealed. Likewise, if you have a situation where sets are of different ages, early sets may be sealed (or partly sealed) and later ones more open. An example is an outcrop of veins abutted or crossed by later joints. These abutting and crossing relations will have a different impact on flow than a bunch of intersecting joints. Maybe in say: "If all the fractures are open, a network with prevalence of I nodes..." This may not be central to the point that you are making in this paper, but it's such a common and misleading logical jump in fracture network studies that the clarification is useful. See the discussion in Forstner and Laubach, 2022, J. Struct. Geol. Also, if the rock itself is porous, even a network that has only I nodes can markedly augment fluid flow (Philip et al., 2005, SPE Res. Eval. Eng.), here length distribution is the key parameter (not connectivity) which just makes your focus on length even more important.

We changed the line as following:

106 (144) In a non-porous rock with all open fractures, a network with a prevalence of I nodes is less connected and thus fluid flow is usually more restrained. Conversely, both Y/T and X nodes increase the connectivity of the network and thus increase both fluid flow and the permeability. However, sometimes this is indeed not the case (e.g. sealed faults and opening-mode fractures) (Forstner and Laubach, 2022). Moreover, if a rock is porous then length becomes the key parameter for controlling fluid flow and thus even a network that has only I nodes can markedly augment fluid flow (Philip et al., 2005).

Line 122: It seems like these values might also be meaningless for 'stochastic modeling'? Do you clarify this in the Discussion?

The discussed values indeed are useful for stochastic modeling both from a statistical and numerical point of view (i.e. DFNs). In the text this was not explained clearly. We provide the following edit to clarify.

125 (171) On the other hand, circular scanline methods offer an unbiased estimate of the mean length, however, being non-parametric, they do not yield neither the distribution type (e.g. normal, exponential, etc.), nor distribution shape parameters (e.g. standard deviation, variance, etc.). This in turn makes the estimate's use quite limited and not apt to downstream statistical modelling applications (such as DFNs).

Lines 129-132: On first read, I found the transitions here confusing. For clarity I think you ought to warn the reader here that you are going to demonstrate the time-length dimensional shift in 3.3. Something like 'Survival analysis is usually used in the time domain. In section 3.3. we show how a time-length dimensional shift is valid. Here we briefly introduce the terms as they are used in the time domain.' These are key lines defining terms. I think they could use some

clarification. What do you mean by 'the even of interest is commonly defined as *death*'? Is a clarifying word missing? The 'event of "x" is'? Or do you need a some more information at the start of the paragraph: "Survival analysis is used to analyze data in which the time until the event is of interest (for example, the time until death in some medical or biological contexts)." This would perhaps be a good point to introduce the idea that you are substituting distance for time?

We changed the text as follows hoping to make it clearer.

128 (176) To solve these problems, we propose to use survival analysis, a specialized field of statistics, developed to deal with censored data. Survival analysis focuses on the analysis of time of occurrence until an event of interest (Kalbfleisch and Prentice, 2002). Although in literature the terms survival times, time-to-failure, or more generally lifetimes (Lawless, 2003) seem to imply that time is the only valid variable, any non-negative continuous variable, such as length, can be valid (Kalbfleisch and Prentice, 2002; Lawless, 2003). The advantage of survival analysis over the methods discussed above is that it considers censored data as a valid datapoint, carrier of the information that the event did not occur up to the censoring time. This is a necessary shift in perspective that allows for an unbiased estimation of statistical models that will be described in the following. We will start by describing the theory of survival analysis in function of time, and then we will show how the same theory can be applied in space, to sets of length or distance measurements.

138 (189) Since this technique is rooted in medical and biological applications, the nomenclature from this type of literature is carried along. The event of interest (for which we measure the time-to-event) is often defined as death, while a loss indicates that the observation has been lost because it was hindered by a secondary event, called a censoring event (Kaplan and Meier, 1958)

Line 133: Which 'length' do you mean here?

Changed with:

143 (194) The event happens after the end of the study period and thus the partial lifetime of the event is observed;

Line 173: 1D or 2D? How does this conversion work?

We did not understand to what this the comment was referred to. If it was referring to Figure 6, indicating that it was not clear what the figure entailed, we expanded the figure caption and text as follows:

Figure 6. Censoring effect on an example of a simple fracture network and corresponding survival diagram. The survival diagram is a 1D representation of the fracture length. On the Y axis the fracture number is indicated and on the X axis the length is represented. Solid lines indicate the actual measured length while dashed lines indicate the possible continuation of

the fracture. Yellow pentagons represent the censoring event given by the intersection of the fracture with the boundary.

Line 190: 'simplest'

194 (255) Changed as suggested

Line 204: 'it has its limitations'

209 (271) Changed as suggested

Line 223: '...that can enable the researcher to obtain an informed...'

225 (298) Changed as: that guides the researcher in an informed

Line 235: 'both figures'

249 (324) Changed as suggested

Line 350: Given the limitations of any spacing statistic, I think it would be worthwhile mentioning here that good field practice with scanlines should be to keep track of the sequence of fracture occurrences, in other words, the spatial arrangement, as you've pointed out in other work (and also Marrett et al. 2018, J Struct Geol). Your analysis here seems like it would be equally apt for spatial arrangement data collection and analysis.

We added a brief mention of this in the text as such

365 (451) It is worth noting that the censoring analysis is a secondary part in the analysis for spacing. Analysing the spatial arrangement of the fractures in the network (such as Marrett et al. 2018 and Bistacchi et.al 2020) is of fundamental importance however, we decided not to include this analysis and focus mainly on censoring to avoid increasing the length of an already dense text.

Line 376: I agree with this way of proceeding re: defining length. Does your method work as well with lengths defined via branches; is there a reason to choose one or the other? Maybe this gets out of scope, but the way you mention it here might make a reader wonder.

Yes, we chose to measure the lengths of the entire segments instead of branches because they entail two different things. Branches offer a useful topological abstraction of the network (making it possible to classify node intersections), but they do not have a real geological or physical meaning. As we defined in section 2, 2D fractures traces are the intersection of discontinuity surfaces with a secondary surface. Branches on the other hand are defined as a segment of a fracture trace between any two nodes (either I-I, I-Y etc..). Considering the geological origin of a

trace, by using branches we would be essentially segmenting fracture planes in smaller sub-planes. This, however, is only an artifact given by the topological definition of a branch and thus the obtained branch length distribution does not carry any real physical meaning.

This discussion, may be a bit out of scope and we tried our best to summarize it in the discussion as follows:

393 (482) Branches offer a useful topological abstraction of the network (making it possible to classify node intersections), however they do not carry a real geological or physical meaning, and thus a distribution obtained by fitting branch-length will have a different meaning compared to one obtained from complete lengths.

Line 388: This is a big claim that length is always underestimated. What if you have a process that produces only short fractures (or even fractures that are shorter than your outcrop size). Hooker et al. 2013, J. Struct. Geol. describes one set (of several) that only contains very short lengths. Maybe some caveats are in order here.

Yes, however we firmly support it and we expanded the discussion to motivate it further:

451 (557) This effect is due to the fact that measured lengths of censored fractures will always be shorter than their true lengths. Thus, by using the first simple approach, the dataset is essentially "polluted" by shorter fractures thus always decreasing the measured mean. The second simple method will also lead to an underestimation of the mean because of the size bias. However, this second method can be less impacted by censoring. For example, if a fracture population has a very small standard deviation (i.e. almost all fractures have the same length) and/or fractures are occurring in an outcrop that is much bigger than the characteristic fracture length, then removing censored values would not have a great impact on the estimation. But, even if small, the underestimation will always be present. Overestimation of the mean length would be possible in these scenarios when we do not consider censoring as independent from the length distribution (for example if only fractures shorter than a certain value are systematically censored). However, this would violate both the core underlying hypothesis of random censoring, and standard geological experience, and thus we do not deem it possible under these imposed limits.

Line 395: 'it'? Maybe 'they are'?

418 (509) they are

Line 405: Although testing this hypothesis is something that people studying fracture lengths in the context of geomorphology ought to consider. Particularly large or open fractures can affect the size, shape, and occurrence of outcrop. See Eppes et al. 2024, Earth Surface Dynamics, doi.org/10.5194/esurf-12-35-2024.

We added this remark in

430 (525) Such discussion highlights that the assumption of independence is difficult to rigorously prove since the true distribution of the length of fractures is not known (we only observe a set of complete and censored data). In some applications (Eppes et al. 2024) this assumption may not hold, and a more in-depth study may be required to prove the independence hypothesis before proceeding. Nonetheless, we believe that it can be safely assumed in geological applications when the appropriate field work and a posteriori analysis are carried out.

Line 409: 'in key of time' is an odd phrase. Check.

437 (533) in function of time

Line 411: 'useless' seems harsh. I'm not convinced this extra remark is needed. Anyway, there may be other parameters (like segmentation) that have similar effects to outcrop size that would benefit from the approach you propose, even if outcrops were arbitrarily large.

439 (535) Removed

Line 445:This assumes that measurements are only caried out at one scale of resolution. But this need not be the case. See Ortega et al. 2006, AAPG Bulletin (for aperture sizes) and Forstner et al. for lengths.

Changed to

484 (581) Due to this limitation, for a constant resolution scale, the modelled length distribution

Line 451: And for some fracture systems, the smaller fractures are more prone to be mineral filled and potentially less obvious features on images. This size/visibility effect can also manifest in the picking of long fractures if the long traces are segmented.

Added as another factor contribution to censoring as follows:

488 (585) Moreover, for some fracture systems, the smaller fractures are more prone to be mineral filled and potentially less obvious features on images. This usually results in fractures that are left uninterpreted even if visible and thus leading to an increase of the real $xmin$ value.

In the example case studies, with such big clear outcrops, can you analyze a small area within the larger area and verify that you are accounting for the censoring correctly?

This is a tricky question that we thought about while writing the paper. It is not easy to see if censoring is correctly accounted for by just subsampling the outcrop (as large as it is). The problem is that essentially, we do not have a controlled environment. First, we are estimating only a limited suite of statistical models and thus we cannot say if the best estimated model is the true underlying model (and in fact we will never be able to tell). So even if for the first case study the lognormal may seem perfectly fitting, we cannot be certain that it is the true underlying statistical model. Moreover, the spatial distribution of the fractures in the outcrop space is also not uniform thus with the same sub area dimension you will have different model estimations depending on the position of the sub area. Thus, we found it difficult to obtain a satisfactory estimate of how well censoring is accounted and how well survival analysis works depending on the censoring percentage. We are relying on the fact that survival analysis has been used and is still being used in countless applications and show that it is working also for lengths.

However, we believe that synthetic experiments can and should be carried out to explore further the effects of censoring, violations of the underlying hypothesis on the final estimation and the overall precision and reliability of survival analysis (we talk about this in 407-435). We decided to not include or explore synthetic results because it would have drastically increased the length of the MS and blurred its focus.

**Sarah Weihmann**

Some comments were related to different aspects of the same paragraphs/lines, in those cases we provide an answer that spans multiple comments.

Line 3: Too vague. What and who's necessities shifted and why now? Who demands parametric solutions?

Line 4: Too vague. Please rephrase "parametric solutions to compare […] all the parameters". Avoid doubling word stems (parameter) and avoid being too general ("all the", "several types").

We applied the suggestions for the lines 3-4 as such:

3 (3) This is due to the fact that, in the past, estimating any type of statistical distribution was difficult and there was no real interest in defining precise parametrical models. However, due to the recent raise in popularity of Digital Outcrop Models (DOMs) and of stochastic Discrete Fracture Networks (DFNs), there is an increasing demand for distribution-based solutions that output a correct estimation of parameters for a given proposed model (e.g. mean and standard deviation)

Line 5: Word order swap: Move "in geological literature" backwards. Work with the "absence of [something]" rather than interrupting this segment.

Line 5: In line 3 present tense was used, now past tense. I recommend present tense here. Further: "These changing requirements" is fine to say once they are specified, else the link is lost (see comment line 4)

Changed as such:

7 (9) This change in demand highlights in geological literature the absence of properly structured theoretical works on this topic.

Line 6: When using "in particular" I recommend using "in general" beforehand to tie the parts together. Again, missing links in storyline.

Line 7: At this point the reader does not necessarily know about right and left censoring, as only later explained in line 133. Either explain here or mention that it is a specific type of censoring that will be explained later in detail. Further: Mark word in italics?

We applied the suggestions for the lines 6-7 as such:

8 (10) Our methodology, presented for the first time in this contribution, represents a powerful alternative to non-parametrical methods, aimed at specifically treating censoring bias and obtain an unbiased trace-length statistical model.

Line: 11: I don't understand "modified" in this context or location. Further, please avoid long sentences; split this sentence into two sentences?

Changed as suggested:

12 (17) We combine a direct visual approach and the calculation of four statistics to quantify how well proposed models reflect the data.

Line 12: How often or on how much data? Maybe place the term "correctly" more prominently; it gets a little lost while this is the main selling point!

Modified as such:

14 (20) We apply survival analysis to correctly estimate statistical parameters of censored length dataset in three different case studies and show the effects of censoring percentage on parametrical estimations that do not use this paradigm.

Line 16: composed of

19 (26) Changed as suggested

Line 19: Replace "Nowadays the increase in" with "Amplified"? Wording sounds dated. Does non-Italian research in DOMs exist? It feels biased. Try being more diverse in reference selection.

Line 21: Grammar: Plural. DOMS allow the extraction of datasets? Try being more exact in wording.

We applied the suggestions for the lines 19-21 as such:

22 (29) The recent increase in computing power and the emergence of new approaches based on Digital Outcrop Models (DOMs) allow for the extraction of large datasets and facilitate the sampling of properties instead of just their estimation (Bistacchi et al., 2015; Tavani et al., 2016; Healy et al., 2017; Thiele et al., 2017; Marrett et al., 2018; Nyberg et al., 2018; Bistacchi et al., 2020; Martinelli et al., 2020; Bistacchi et al., 2022; Mittempergher and Bistacchi, 2022; Storti et al., 2022).

Line 32: "only indirect geophysical methods may provide truly 3D datasets". Truly as adverb to provide? Or true as adjective to 3D datasets? Change word order or grammar. Further, is there a better way of saying it?

38 (46) Changed "truly" with "complete"

Line 34: "a rich literature" sounds odd. Try "vast research conducted" or else.

Line 34: Please don't start sentences with "Because of". Try "Due to" or "Given" or else. Please follow academic writing guides.

Line 35: "the 2D lines of intersection of 3D [...] surfaces with the outcrop surface, or with topography" is too complicated. Please make sure to keep sentences short and clear.

We applied the suggestions for the lines 34-35 as such:

40 (50) Therefore, a vast body of research focuses on the characterization of properties of discontinuity traces or lineaments, i.e. the 2D intersections of 3D discontinuity surfaces with

the outcrop surface, or with topography (Dershowitz and Herda, 1992; Bonnet et al., 2001; Manzocchi, 2002; Bistacchi et al., 2011; Sanderson and Nixon, 2015; Bistacchi et al.; 2020; Martinelli et al., 2020; Storti et al., 2022).

Line 41: "the Digital Outcrop approach". Is this a standardised method? It has not been mentioned in the text before. It is also not explained in more detail in the following sentence as the reader might expect. Please link the sentences more carefully and guide the reader better. Please consult academic writing guides.

The wording may confuse the reader since it is not clear that we mean that DOMs are the tools used and not the methods for defining unbiased lengths. We rephrased as such:

44 (58) In this contribution we focus on the problem of defining accurate and unbiased length (or height) distributions based on data collected on DOMs

Line 43: don't capitalize "authors"

47 (61) Changed as suggested

Line 44: "consists of"

48 (63) Changed as suggested

Line 47: Why reference here and not at the end of the sentence?

51 (65) Modified

Lines 50-52: Please don't put words in bold. They have no meaning individually here and the reader is able to read text just so. They stand out too much in light of the text here.

52-54 (69-71) Removed bold

Line 53: Please avoid disclosing information in brackets. Instead, convert them to separate sentences.

We have changed by removing the content in brackets

51 (67) Thanks to its simple implementation in the field, this technique is widely used, however it has an important limitation: lineament lengths are never directly measured. For this reason, analysis carried out with the circular scanline method yield estimates of mean length values but are unable to provide a complete characterization of the lineament length distribution and therefore lack any real statistical significance.

Line 66: time spans. Also, please do not put in bold.

71 (89) Changed

Line 67: length measurements

Line 69 "that is the main topic of this contribution" is a separate, full sentence. Do not add this to the previous sentence. Please check academic writing guides. What is physical measuring? Please clarify.

72 (91) Measuring length is straightforward with dedicated code or with a simple GIS software however, applying survival analysis and fitting robust unbiased parametric statistical

distributions, needs a more detailed treatment. As the main topic of this contribution, we propose to adapt survival/reliability analysis techniques to correctly account for censored lengths and estimate robust trace length distributions derived from DOMs.

Line 72: Avoid "second objective" if there has been no "first objective". Please re-read academic writing guides. Always stay consistent.

Line 73: These hypotheses come as a surprise, as they have neither been introduced nor are they explained here. Please adjust.

75 (96) Furthermore, we define a quantitative methodology to select the most representative estimated statistical model (i.e. parametric distribution) from a list of proposed models.

Line 79: Language is imprecise. Joints are never "empty". There will always be fluids/gases if not solids. Better: lacking mineral filling, or else

Previously modified with the comments provided by Laubach (84 (110))

Line 85-86: Avoid making multiple points in one sentence. It exhausts the reader too quickly.

We have split the sentence as follows:

87 (114) Although three dimensional by nature, most of the times discontinuities are mapped as 2D lineaments or traces over a surface. These lineaments are the intersections of such discontinuities with an exposed surface, such as the topography, an outcrop, a borehole or a sample.

Line 88-90: Split in two sentences please.

Modified as such:

89 (119) In this work, following a common usage in outcrop studies, the term fracture is used as a generic term to indicate any type of discontinuity trace. Fractures with the same formation age, kinematics and orientation, can be grouped into families or sets. Multiple fracture sets present within an area, form a fracture network or system (Davis et al., 2012)

Line 91: Does the sampling area reach from a thin section to a satellite image? Language is not precise enough. Please improve.

Line 92: Please do not put new points in brackets. Either make them a new sentence, if important, or leave them completely out.

Line 93: Why "boundary nodes" in bold if not "nodes and branches" (line 98) in bold as well? Please stay consistent. Generally, avoid bold style.

Changed suggestions for lines 91-93 as follows:

93 (124) Boundaries are the limits of the sampling area, at scales from thin sections to satellite images, within which the sampled fracture traces are assumed to be complete.

Line 94: Please rearrange the sentence order. Think about what point is trying to be made and move it either to the front or end of the sentence. Please check academic writing guides for this.

Line 94: "impossible" is too emotional. Use "not feasible" or else.

Line 95: Never use "..." in an academic manuscript.

Changed the text following suggestions for lines 94-95:

95 (129) Ideally, boundaries are strictly convex, however this is often not the case. Boundaries often show tight bends, coves and holes and the final shapes are mostly controlled by localized alteration, anthropogenic activity, vegetation etc. (Fig. 1B)

Line 100: If it is not a direct quote there is no need for a page reference

98 (133) We removed the citation to better accommodate the text changes.

Line 106/107: Why future tense and not present tense? In line 110 present tense is used. Please stay consistent.

Previously modified with the comments provided by Laubach (106 (144))

Figure 1: Maybe mark the "hole" with red hatching? This makes it easier to find the "hole" quickly and follow the explanation.

Changed as suggested

Figure 2: "Topology" seems to be spelled with an odd character. Number 8 in the right-hand picture seems dislocated.

Fixed

Line 114/115: The start of the sentence and paragraph is poor. Try "There is increased necessity for estimating parameters of statistical distribution in length datasets". Before "however" there should be a full stop. First person plural should be avoided.

Changed as such:

120 (162) Having a length dataset, there is usually the necessity of estimating the parameters of one or multiple statistical distributions. When doing so, censoring is inevitable as the area within which measures are carried out will always be limited.

Avoid putting all but one word of a sentence in bold. It looks accidental. Maybe introduce the question with a colon or just don't make it bold.

Changed as such

121 (167) Then how can we carry out an unbiased estimation of such parameters?

Line 119: "lines" = line

125 (171) Changed as indicated

Line 120: Do not write "on the other hand" if there is no "on the one hand".

Changed by adding :

122 (167) On the one hand

Line 122: "almost completely meaningless and useless" is too emotional. Use "impractical" or else.

Changed as:

127 (175) This in turn makes the estimate's use quite limited and not apt to downstream statistical modelling applications (such as DFNs).

Line 124-126: This sentence makes no sense to me. Please rewrite.

We expanded this concept as follows:

132 (180) The advantage of survival analysis over the other methods discussed above is that it considers censored data as a valid datapoint, carrier of the information that the event did not occur up to the censoring time. This is a necessary shift in perspective that allows for an unbiased estimation of statistical models that will be described in the following.

Lines 133-140: Always start with a capitalized letter after a colon.

144 (194) Changed as suggested

Line 134: Why first-person plural here? Please avoid. Use passive voice. Maybe use: "the event happens after the end of the study period and thus the length of the event is partially observed".

Changed to:

144 (194) The event happens after the end of the study period and thus the partial lifetime of the event is observed;

Lines 142-145: Avoid writing in bold.

151 (204) Removed bold

Line 145: time to failure or time-to-failure? Keep spelling constant. Also, this term was only mentioned in an example and inside brackets – explain this more in detail instead.

Changed to:

154 (208) lifetime

Line 150: I understand "complement" as a supplement or accessory. Is "inverse" meant?

Changed as such:

159 (212) The survival function is often called complementary cumulative function, since it adds to 1 with the cumulative distribution function (**CDF**)

Line 166: "non-negative continuous variable" is not defined and is not the opposite to "valid variable" yet seems to be the "central point of this work". The term needs more introduction if it holds such importance.

This concept was moved in the introductory paragraph of the chapter. Thus we changed as such:

130 (178) Although in literature the terms survival times, time-to-failure, or more generally lifetimes (Lawless, 2003) seem to imply that time is the only valid variable, any non-negative continuous variable, such as length, can be valid (Kalbfleisch and Prentice, 2002; Lawless, 2003).

...

175 (229) Since length is, as time, a non-negative continuous variable, it is theoretically possible to apply survival analysis techniques to length datasets by considering (Fig. 6):

Line 170: Why highlight a verb if else only nouns are put in bold? Stay consistent by changing wording or marking.

Changed as such

177 (233) the complete fracture trace length analogous to the lifetime;

Line 174: What are "the definitions of the different types of censoring"? They are not mentioned. The bullet point list mentions considerations. Why are definitions referred that are not mentioned in this subchapter?

We provided a link to the section where the types of censoring are described as such:

181 (237) By applying the definitions of the different types of censoring (described in 3.1) to our specific application ...

Line 176: Why are sources of fracture genesis mentioned here in this subchapter? Is this not a topic for the introduction?

The sources in parentheses were related to secondary events (the boundary) not to fracture genesis. We understand the confusion caused by the wrong placement of the parenthesis:

183 (239) Moreover, censoring is non-informative since the boundary is the product of secondary events (i.e. alteration, erosion, debris covering parts of the outcrop, vegetation, human activity, etc.) that occur after the fracture genesis and thus do not inform the occurrence of the event (see the discussion for a more in-depth analysis)

Line 189: If a colon is used to present the main objective, avoid listing side points as bullet points. The message gets lost, and the reader is confused.

Changed as follows:

191 (247) Given a statistical parametric model with density g(x, θ) (i.e. an assumed theoretical distribution) and a sample x of size n, the main objective of MLE is to estimate the parameters $\hat{\theta}$ such that the observations x are the most likely under g(x, $\hat{\theta}$) (Burnham and Anderson, 2002, 2004; Karim and Islam, 2019)

Line 190: simples to simple

Previously modified with the comments provided by Laubach (194 (255))

Line 196: Avoid putting words in bold

200 (261) Removed bold

Line 209: uncertainty to uncertainties

214 (277) uncertainties

Line 212: What are natural questions? Please clarify.

Line 213: Changing several short simple questions to a long complicated question is not "reducing". Starting a question with a side sentence is not recommended. Structure needs to be clearer.

Changed as follows to fix both comments :

218 (285) With a series of fitted models it is important to understand which model better represents the data. In literature this is usually archived using a specific type of null hypothesis tests defined as goodness-of-fit (GoF) test (i.e. Storti et al., 2022).

Line 216 ff: Do not use first person plural. Instead: "These types of tests…"

219 (285) Changed as indicated and expanded to have a more precise definition:

These types of tests, in general, do not identify the best fitting distribution among a set of possible distributions. A GoF test takes a distribution L and the sample data, and tells if the model L is not plausible, that is, if the probability of drawing such a set of data, from a population of distribution L, is too small. This means that more than one distribution may be deemed plausible and the conclusion is strong only when a distribution is an unlikely model for the data.

Line 224: "Sensible" sounds highly subjective if not explained how this is defined.

Line 256: "models deemed reasonable by the researcher" again is very subjective. Can this be made more objective?

We added the following to address the above comments:

226 (299) By sensible candidates it is intended those models that make physical sense for the case study. For example, for fracture lengths, a power law can be considered a sensible candidate since it describes an observed pattern of fracture self-similarity. A lognormal is also an equally sensible candidate because of the effect of truncation. Conversely a normal distribution is not very sensible since values can be negative. Thus, for a normal model, either the average length is very high and standard deviation very low or a truncated normal should be tested. In the case studies section, we will cover different models purposefully adding non-sensible models (such as a standard normal) to showcase the model selection workflow.

Line 226: Do not start a sentence with rather meaningless introduction words. The reader's interest is immediately lost. Instead start with the subject "Probability Integral Transform is [...]"

Lines 228-231: Do not name the conditions (list) if the transformation statement is promised after the colon. Instead write line 231 first after the colon. The list gives too much importance to the conditions and limits the focus on the actual definition.

Changed as following to address both comments:

234 (307) The Probability Integral Transform (PIT) is a well-known transformation of continuous distributions which converts random variables with continuous distribution to standard uniform distributed random variables (Fisher, 1990). Indeed given:

....

Line 233: Split sentences after "(Fig. 7A)". One thought per sentence only.

247 (322) Added a full stop after fig7A

Line 248: likelihood of...

262 (339) the likelihood of the model (L)

Line 295: Again, important messages need to be at the start or end of the paragraph. This is not the case here. Make sure "sensible guided choice" is put in the end.

Changed as follows:

307 (385) By comparing multiple rankings, even if the calculated distances are ranked in differently, it is still possible to make a sensible guided choice, for example by using the PIT representation together with the mean ranking position or following a specific type of distance.

Figure 3 and line 136: "observations intervals" = observation intervals". A figure should always speak for itself. This figure does not make enough sense by itself and needs simplification. Reduce the number of colours, avoid unnecessary and unlisted abbreviation (e.g. start and end), make sure colours can be distinguished (e.g. black vs. grey; tightly-dashed vs. line), explain question marks, match thickness of lines in image and legend, standardise spacing behind "Complete" and "Interval", standardise font size for y-axis title, y-axis is not an axis, etc.

Figure 4: standardise axis titles font sizes, match image and legend line thickness, explain "C" in figure, clarify definition for partial length, use "s" and "r" in the image if explained within the legend, A and B should be on the top left side of each image, full-stop missing in figure description.

Figure 5: Standardise font size for A and B, make box clear unless figure B is red, increase all axis title and value font sizes, remove title for figure or increase font size, define axis title x.

Figure 6: Match axis title font sizes to legend font size, Left: Match legend to image. Right: Match line thicknesses, clarify "n." on y-axis: title number would be abbreviated "no.", why clarify unit of length but not item (fracture length)?

Figure 7: The numbering (A, B, C) would normally occur on the left-hand side of the sub-figures. The text appears rather small and might be increased – however the message is clear.

Changed all figures as suggested except for letter position to avoid obstructing the Y axis numbers.

Line 297: "all the discussed theory" to "the discussed theory"

311 (390) the discussed theory

Line 298: In the sentence before it is "case studies" so one needs to introduce singular first: change "one" to "case study"

312 (391) the first case study

Line 307 (caption Fig. 8): Try to mirror sentences that belong together. If starting with "Pictured on the left…" continue similarly to guide the reader. Don't say "On the right it is represented" but try "Pictured on the right" or similar.

Caption Fig.8  Pictured on the right, a subarea of the quarry with the boundary, fracture traces and boundary nodes (yellow pentagons)

Line 327: "Weibull seem" to "Weibull model seems"; "as" to "than"

Changed as follows:

339 (419) While the Koziol and Green distance favours the gamma model, the Anderson-Darling distance favours the Weibull model.

Line 328: Again, the last sentence has little value in the paragraph ("occupy the last two positions"). Maybe better: "rank lowest in comparison of the distances/models" or "are least representative".

Changed both as following:

340 (422) Finally, the power law and normal models rank lowest in comparison to the other models indicating that they are less able to represent the dataset.

Line 336, 337: Why past tense here when nowhere else?

Changed as following:

349 (432) This leads to …

…

350 (432) The analysed fractures are subdivided …

Line 341. Do not start a paragraph and/or sentence with the least important information ("In Fig. 12").

354 (437) The different estimated models are represented for both sets in Fig. 12.

Line 341/342: Can this sentence be smoother?

354 (438) In both cases the lognormal model, overall, better fits the data

Line 346: replace "afterwards" as it is too figuratively. Try "at greater lengths" or else.

358 (443) for longer length values

Line 349: "For the other models, looking at the mean rank value helps in understanding the final ranking showing that the gamma distribution is ranked lower than the exponential and the Weibull (at the second and third place respectively)" reads very clunkily. Try and rephrase.

Changed as such:

360 (444) For the other models, the mean rank value can be used to assign a final ranking. The mean ranks show that the gamma distribution is less representative than both the exponential and the Weibull (respectively the second and third most representative models).

Line 350: Case study 1 and 2 are locations whereas case study 3 is a topic. Maybe make that clearer by expanding the title, e.g. "Spacing analysis"

Changed subsection 4.3 title as suggested

Line 351: Do not start a paragraph or sentence with minor information like here. Start with "Survival analysis can be used…"

We have changed as suggested:

364 (449) Survival analysis can be used also to analyse the spacing length distribution for each fracture set. Thus, the same workflow was applied to spacing measurements of both S1 and S2 of the Colle Salza dataset.

Line 366: Start new sentence with/at "Thus"

383 (469) Changed as indicated

Line 367: Add "model" to end of first sentence.

383 (469) Added model at the end

Line 367: "bad" to "poor".

Line 368: Don't use "place". Try "rank" or else.

383 (469) The poor fit of the lognormal model is also confirmed by the other distances that rank the lognormal lower than the Weibull and gamma models (ranking first and second respectively).

Line 370: Do not use "on the other hand" if you haven't used "on the one hand". Mind structure and consistency please.

386 (473) Conversely, the lognormal model is clearly the most representative for set 2 (Fig 15B).

Line 371: What does "converging all at the first place" mean?

Line 370/371: Repeated use of word "quite"

Changed as follows to accommodate both comments:

387 (474) The PIT plot showing a quite linear behaviour and all the calculated distances are ranking first.

Figure 8: Text is very small. "Yellow pentagons" can't be identified at this scale. Second scale in left figure should be inside the zoom window.

Figure 9: Remove title or make bigger, reference length double units name standardized/ true.

Figure 10: Title is too large, table is too small. Numbers and axis titles are small and hard to read. I suggest putting numbering of subplots (A-D) on the left-hand side of each image. I would get rid of all articles to make the figure description consistent. Overall title: "plots"?

Figure 11: Numbers and words need larger font. Insert scale of subplot left inside subplot box.

Figure 12: Increase font size of title and axis title. A and B should be on left-hand side of figure and potentially smaller (compare with other figures). The text could read better. Try "PIT visualization for the proposed length models is shown for Set 1 (A) and Set 2 (B) of the Colle Salza dataset. The red line represents the reference U(0,1); the closer a model's line is to this reference, the more representative the model. Among the models, the lognormal distribution demonstrates the closest fit to the reference line in both sets, although its fit is inferior to that observed in the first case study. Across both sets, all estimated models exhibit less linearity, with notable underestimation between 0.34 m and 1.5 m in Set 1 (A) and between 0.44 m and 2.57 m in Set 2 (B)." or similar.

Figure 13: Reduce title font, increase table font, change numbering A-D to left-side please. Increase axis description. Here, all articles are kept consistent in the description. This is better than in Fig. 10.

Figure 14: Reduce A-C numbering (compare with other figures). Increase legend. Explain all lines in legend. Purple colour hard to see. Stay consistent in your phrasing: Delete "Shows".

Figure 15: Reduce numbering letter A and B and title, compare to other figures. Increase y-axis and clarify. Please correct figure description by the advice given above.

Figure 16: Please adjust sizes referring to similar figures and comments above.

Changed all figures as suggested except for letter position to avoid obstructing the Y axis numbers.

Table 2: "indicating a worst fit in respect of the exponential and Weibull distributions positioned in second and third place respectively" reads very clunkily. "a worst fit" does not exist and "places" are only handed out at races. Try and rephrase.

Changed table 2 caption as follows:

Models' distance and rank tables for both Colle Salza sets. The closer to 0 the better. For this length dataset, the lognormal is the most representative of the data for all the different distances. Using the mean rank, the lognormal is followed by the exponential, Weibull and

gamma distributions. As with the first case study the Power law and normal distribution are the less representative.

Line 374: Please don't put information in brackets

Changed as follows:

391 (478) With this work we delineate a robust statistical framework to quantitively analyse and statistically model fracture trace length and spacing.

Line 384/385: Repeated use of word "useful"

Changed as:

405 (496) Furthermore, having an unbiased statical model can be also extremely important for engineering applications. For example, Pahl (1981) states at the end of his work that in geomechanical application it could be useful to know the longest trace likely to occur in a group of traces.

Line 407: Please only mention a "second point" if a "first point" was explicitly named (line 378 names a "crucial" point; maybe write "The first and crucial point"?

Added the following:

401 (490) The first and crucial point

Figure 17: Increase/ add axis titles and number. Increase titles (compared to numbering (A-D)). Increase legend size.

Changed figure as suggested

**David Healy**

Line 21: 'allows the extraction of large datasets and facilitates the measurement of properties' – these are still just samples from the population though; they are not the 'right' answer.

23 (32) Changed "measurement" with "sampling"

Line 50: note that FracPaQ does not in fact use the mean length statistic from these measures, just Intensity and Density; length statistics are calculated directly from the sample lengths. In addition, FracPaQ employs MLE methods to estimate optimum length distributions, with a Goodness of Fit approach. The work of Rizzo et al., (Rizzo, R.E., Healy, D. and De Siena, L., 2017. Benefits of maximum likelihood estimators for fracture attribute analysis: Implications for permeability and up-scaling. Journal of Structural Geology, 95, pp.17-31.) is not cited here, and it needs to be. As the current text gives an incorrect impression of FracPaQ functionality, I respectfully ask for clarification on these points.

Line 383: again, as above length stats estimation in FracPaQ does not use circular scanlines; we use the mean and standard deviation of the sample data and, optionally, MLE. Please correct this misleading statement.

We again sincerely apologize for the misleading statements. We correct this and all the other instances as follows (better viewed in the diff file)

46 (60) The correction of these biases has been thoroughly researched, and the standard solution currently adopted by many authors is based on circular scanlines (Mauldon, 1998; Zhang and Einstein, 1998; Mauldon et al., 2001; Rohrbaugh et al., 2002).

51 (67) Thanks to its simple implementation in the field, this technique is widely used, however it has an important limitation: lineament lengths are never directly measured

59 (77) These limitations have been mostly overcome by modern characterization methods and thus new tools and techniques based on Maximum Likelihood Estimation such as FracPaQ (Healy et al., 2017; Rizzo et al., 2017) have been developed, enabling researchers to apply quantitative statistical inference on dense digital datasets.

404 (494) Due to this, non-parametrical methods such as those proposed by Mauldon et al. (2001) are unfit since they are not linked to any model.

409 (500) Modern alternatives that use a simple implementation of MLE such as FracPaQ (Healy et al., 2017) are a good step forward, however the censoring bias is still present and thus the results are biased, always underestimating length.

Line 118: 'avoided at all costs' is a bit too dramatic; delete.

124 (169) Changed to: should be avoided

Line 123: 'completely meaningless' – again, too strong; with no other alternative, it can be a useful estimate, albeit limited.

Previously modified with the comments provided by Weihmann (127 (175))

Line 405: not sure this statement is true. Many outcrops are bounded by fractures; thus, the modern day process that has defined the boundary HAS been influenced by the geological structure and fabric of the rock mass

We understand that the statement as is written is confusing and not necessarily true. We have rewritten this part to include a clearer explanation of independence (including a new part suggested by Laubach's review):

423 (514) To correctly classify censoring as random, we must assume independence between the censoring and length distribution. By "independence" it is intended that the mechanisms behind the generation of a fracture length distribution is different from the mechanisms that censors such lengths. The boundary, which represents censoring, is usually the product of secondary events that occur after fracture genesis (i.e. alteration, debris hiding part of the outcrop, vegetation, human activity, etc.). Thus, albeit it is often the case that such events are controlled by preexisting structures, the physical processes that caused censoring are not the same that generated the fracture set and thus the original length distribution. This leads to an important implicit caveat where the measured lengths must be related only to the mechanism that we are interested in modelling, for example lengths that are surely linked to tectonics and no other secondary events. Such discussion highlights that the assumption of independence is difficult to rigorously prove since the true distribution of the length of fractures is not known (we only observe a set of complete and censored data). In some applications (Eppes et al. 2024) this assumption may not hold, and a more in-depth study may be required to prove the independence hypothesis before proceeding. Nonetheless, we believe that it can be safely assumed in geological applications when the appropriate field work and a posteriori analysis are carried out.

Line 470 – 'proper' – replace with 'better'.

511 (608) better

Line 485 – regarding DFNs (and elsewhere in the ms); there are other approaches to modelling fractured rock volumes, for example effective methods and tensorial approximations. It would

be better to mention and acknowledge these alternatives. DFNs are just one approach, among many.

We added the following lines in the introduction:

33 (42) However, it is worth noting that DFNs are not the only viable approach to model fractured rock volumes. Other methods such as tensorial approximations (Suzuki et al 1998, Brown and Bruhn 1998) based on the crack tensor measure (Oda 1989) are also used (Healy et al 2017 ) and, with details depending on the implementation, require similar parameters in input.

Fig 5, 9, 12, 15, 17, – make the axis labels (numbers and text) bigger relative to the figure; hard to read.

Changed also following the suggestion from Weihmann's review.

**Independent additions**

We have added a better explanation of GoF tests from line 219 (287):

A GoF test takes a distribution L and the sample data, and tells if the model L is not plausible, that is, if the probability of drawing such a set of data, from a population of distribution L, is too small. This means that more than one distribution may be deemed plausible and the conclusion is strong only when a distribution is an unlikely model for the data. Moreover, GoF tests usually have underlying assumptions that if ignored undermine the test's accuracy (Storti et al., 2022). For example the Kolmogorov-Smirnov test (Kolmogorov, 1933) is biased if the parameters of the tested distribution are estimated from the data (Bistacchi et al., 2020).

We have added a better explanation of PIT and PIT plots tests from line 234 (307):

The Probability Integral Transform (PIT) is a well-known transformation of continuous distributions which converts random variables with continuous distribution to standard uniform distributed random variables (Fisher, 1990). Indeed, given:

1. A random variable Y;
2. $F_Y$ the CDF of Y;
3. $Z = F_Y(Y)$ is the PIT of Y and is a standard uniform random variable.

This means that, given a random sample X, sampled from a population with distribution Y, the transformed sample $Z = F_Y(X)$ has, with large probability, an empirical distribution which is close to the standard uniform distribution. Clearly, if we transform X with another distribution W, which is not the true distribution of the population, then $F_W(X)$ is not uniformly distributed. This observation provides a visual tool to compare different fits: indeed, suppose that sample X might have been sampled from a population of distribution Y or W. We may compute the two

transformed samples $Z_1 = F_Y(X)$ and $Z_2 = F_W(X)$: if X was drawn from distribution Y, then the empirical distribution of $Z_1$ is likely to be close to the standard uniform, while if the true distribution was W, then we expect that $Z_2$, instead of $Z_1$, would be close to the standard uniform.

And at line 252 (329)

Values in the PIT plot that are below the diagonal indicate an overestimation of the model's CDF in respect to the empirical cumulative. Conversely values that are above the diagonal will result in an underestimation of the model's CDF.

We fixed a mistake in formula 6 at line 260 (237)

And generalized the concept at line 263 (340):

This formulation outputs a real number (either positive or negative) which should be small as the distance between the true population and the model decreases